



# Synthesis of surface snowfall rates and radar-observed storm structures in 10+ years of Northeast US winter storms

Laura M. Tomkins[1], Sandra E. Yuter[1,2], Matthew A. Miller[2], Mariko Oue[3], and Charles N. Helms[4,5]

[1]Center for Geospatial Analytics, North Carolina State University, Raleigh, NC, 27695, USA
[2]Department of Marine, Earth, and Atmospheric Science, North Carolina State University, Raleigh, NC, 27695, USA
[3]School of Marine and Atmospheric Sciences, Stony Brook University, State University of New York, Stony Brook, NY, 11794, USA
[4]Earth System Science Interdisciplinary Center, University of Maryland, College Park, College Park, MD, 20740, USA
[5]NASA Goddard Space Flight Center, Greenbelt, MD, 20771, USA

**Correspondence:** Laura M. Tomkins (lauramtomkins@gmail.com)

**Abstract.**

Winter storms can cause significant societal impacts in the densely-populated regions of the Northeast United States. Mesoscale snow bands embedded within winter storms are often the main focus of snowfall forecasts and analyses. This study investigates the relationship between observed surface snowfall rates and local enhancements in radar reflectivity (i.e. mesoscale snow bands) using data from 264 storm days over 11 winter seasons (2012-2023). We compare hourly surface snowfall rates obtained by ASOS weather stations with the area × time fractions of locally-enhanced reflectivity features and of all echo passing over the 25 km radius vicinity of the surface observation. Our analysis focuses on non-orographic snow storms with surface winds $< 5 \, \mathrm{m \, s^{-1}}$.

Our findings show that most of the time snow rates are low (75% of hours had liquid equivalent snow rates less than $1 \, \mathrm{mm \, hr^{-1}}$). Heavy snow rates ($> 2.5 \, \mathrm{mm \, hr^{-1}}$ liquid equivalent) are rare ($< 4\%$ of observations). When enhanced reflectivity features pass over a location, only 1 out of 4 hours have heavy surface snow rates. High spatial resolution vertical cross sections from airborne radar obtained during the NASA IMPACTS field campaign and rapid update RHIs from ground-based radar demonstrate that enhanced reflectivity features in snow aloft usually lack the vertical column continuity characteristic of reflectivity structures in rain. Ice streamers with higher reflectivities are tilted and smeared on their way to the surface as their constituent snow particles are dispersed laterally by the horizontal winds within the storm.

## 1 Introduction

The intersection of high population density and winter weather in the Northeast United States yields more frequent disruptive societal impacts as compared to winter weather in other US regions (Kocin and Uccellini, 2004; Novak et al., 2023; Guarino and Firestine, 2010). This geographic region encompasses the states of Pennsylvania, New Jersey, New York, Connecticut, Rhode Island, Massachusetts, Vermont, New Hampshire, and Maine and includes the urban corridor spanning the cities of Philadelphia, New York, and Boston. The geography includes portions of the Appalachian Highlands and Atlantic Coastal





Plain (Fenneman, 1928). In general, storms in the region have negligible to weak orographic forcing, excepting a few areas with higher peaks.

Much previous work on diagnosing heavy snow rates in northeast US winter storms has focused on mesoscale *snow bands*, elongated features of enhanced radar reflectivity observed by operational scanning radars. Snow bands that are longer than 200 km are called primary bands and are associated with strong frontogenesis at low- and mid-levels in the storm (Novak et al., 2004, 2010; Ganetis et al., 2018; Kenyon et al., 2020; Baxter and Schumacher, 2017). Bands that are shorter than 200 km and typically occur in groups are known as multi-bands. Previous work related to winter storms in the northeast US has often focused on primary bands and understanding their associated physical mechanisms (e.g. Novak et al., 2004, 2008, 2009, 2010; Novak and Colle, 2012; Kenyon et al., 2020; Stark et al., 2013). The relationship between multi-bands and frontogenesis is not clear as multi-bands are found in a wide range of frontogenesis environments including negative values (frontolysis) (Ganetis et al., 2018; Nicosia and Grumm, 1999; Connelly and Colle, 2019). The physical mechanisms and relative importance of different types of instabilities (potential, conditional, shear, conditional symmetric) to multi-band production is a topic of active research (e.g. Shields et al., 1991; Ganetis et al., 2018; Leonardo and Colle, 2024; Varcie et al., 2022; Zaremba et al., 2024). Case studies of northeast winter storms have usually focused on extreme events with high snowfall accumulations and strong frontogenesis (e.g. Picca et al., 2014; Varcie et al., 2022; Ganetis and Colle, 2015; Novak et al., 2008; Han et al., 2007; Colle et al., 2014; Clark et al., 2002; Lackmann and Thompson, 2019). But the relative prominence of different physical processes in an extreme event may not be analogous to that for a more typical, weaker winter storm. The practice of generalizing that extreme events are representative of typical events implies that extreme events are a clearer, more "pure" representation of the key physical processes rather than a low probability juxtaposition of circumstances that amplify snowfall. The assumption of representativeness of extreme events for more typical winter storms needs to be evaluated with evidence from a large sample size.

We use a NWS NEXRAD radar and ASOS data from 264 snow storm days in the northeast US to determine if the relationships illustrated in outlier case studies between radar observed snow bands and heavy surface snowfall withstand scrutiny with a large data set. We also examine recent field campaign observations of the detailed vertical structures within winter storms from airborne radars deployed during the recent NASA Investigation of Microphysics and Precipitation for Atlantic Coast-Threatening Snowstorms (IMPACTS) field campaign (McMurdie et al., 2022) and from ground-based radars at Stonybrook University (KASPR; Oue et al., 2017).

A complicating factor in the analysis of winter storms using weather radar observations is the interpretation of radar reflectivity in snow. In warm-season precipitation systems it is reasonable to deduce that stronger locally-enhanced radar reflectivity features are associated with higher rain rates at the surface. However, increases in radar reflectivity in snow do not necessarily equate to increases in mass per unit volume (Table 1). Aggregation and partial melting increase the radar reflectivity but do not change the mass per unit volume which yields ambiguities in the physical interpretation of radar reflectivity in snow that are not present for rain. Snow falls slowly ($\sim 1$ m s$^{-1}$) compared to rain ($\sim 5$ m s$^{-1}$) so it is more easily transported by the horizontal winds of $\geq 10$ m s$^{-1}$. Unlike convective cells in warm season precipitation, in snow storms vertical columns of enhanced reflectivity extending several km from mid-levels to the surface are rare.





Coincident observations of ice water content (IWC) and radar reflectivity collected during research flights as part of the Seeded and Natural Orographic Wintertime Clouds: the Idaho Experiment (SNOWIE) field campaign in the mountains of Idaho (Tessendorf et al., 2019) describe the wide variation between IWC and reflectivity in ice clouds and light snow (Zaremba

et al., 2023) (Fig. 1). These IWC and Z data for surface snow producing clouds over the mountains of Idaho do not include any partially melted particles but may include aggregates. For reflectivities $> 0\ \mathrm{dBZ}$ which usually contain some precipitation-size falling ice particles, IWC generally increases as Z increases but given the spread of the observed values, it is difficult to quantify volumetric ice mass as a function of radar reflectivity better than a factor of 2.

The overarching goal of this study is to explore and understand the relationships between surface snow rates and locally-

enhanced reflectivity features (i.e. snow bands) in winter storms. Data and Methods used in this study are discussed in Sec. 2. Section 3 presents the results of this study. Conclusions are described in Sec. 4. The key finding is that snow bands within northeast US snow storms (which exclude orographic and lake effect snow storms) are rarely associated with heavy surface snow rates.

## 2    Data and Methods

### 2.1    10+ year Winter Storm Dataset

We use observations from 264 storm days in the northeast US from over 10 years (2012-2023) to ensure we are studying the relationships over a large representative sample. Following the methodologies of Hoban (2016) and Ganetis et al. (2018), we define a winter storm as any date within the months of October to March for the years 2012-2023 where at least $25.4\ \mathrm{mm}$ (1 inch) of snow depth was reported over a 24-hour period at at least two out of 14 stations shown in Fig. 2b. We used a threshold of

$25.4\ \mathrm{mm}$ to include a wide range of storms in our analysis and not only those that produce a large snow accumulation. The daily data at each station was gathered from the Global Historical Climatology Network daily (GHCNd) database (Menne et al., 2012). We selected the subset of 14 stations to focus on storms that impacted the more densely-populated regions of the northeast US and to minimize the inclusion of storms that were predominantly lake-effect snow events. The years 2012-2023 were chosen since 2012 is the start of the period when dual-polarization radar data are available from US NWS radars. Use of

dual-polarization products allows us to remove melting and mixed precipitation observations, often confused for heavy snow, from our analysis (discussed in Sec. 2.1.3). The full list of storms are tabulated in Table S1 in the Supplementary Material.

### 2.1.1    Hourly surface station observations

Hourly observations from 29 ASOS stations in the northeast US are used to quantify the liquid equivalent snowfall rates during each winter storm day (Fig. 2). The hourly snowfall rates represent hourly liquid equivalent *accumulation* and are not

instantaneous snow rates. ASOS data are obtained from the online MADIS database (https://madis-data.ncep.noaa.gov/index. shtml). Measuring snow in rain gauges at ASOS stations can be challenging as snow is easily blown sideways by the wind and doesn't always make it into the gauge (Rasmussen et al., 2012). To ensure we are getting the best measurements, we are only





using ASOS stations equipped with all-weather precipitation accumulation gauges (AWPAG) as these gauges are more skilled at measuring liquid equivalent snow accumulation than other types of gauges (Martinaitis et al., 2015). The AWPAG sensors

are equipped with Tretyakov or double Alter style shields, which are more accurate at measuring frozen precipitation than gauges with no shields (Rasmussen et al., 2012). AWPAG sensors do not have a heated rim and thus are subject to capping, although it is difficult to estimate how often this occurs in our dataset. If capping does occur, no snow would be reported for an hour. To further ensure that we are using reliable observations, only quality-controlled observations when the wind speed is $< 5 \mathrm{~m~s}^{-1}$ are used in our analysis (Rasmussen et al., 2012). The collection efficiency of frozen precipitation is 1.0 at a wind

speed of $0 \mathrm{~m~s}^{-1}$ and drops to 0.25 at a wind speed of $6 \mathrm{~m~s}^{-1}$ for double Alter-shielded gauges which is why we chose a threshold of $5 \mathrm{~m~s}^{-1}$ (Rasmussen et al., 2012). The wind speed threshold removes $\sim$45% of hourly snow observations. The distribution of removed measurements is relatively uniform over the entire range of precipitation rate observations (see Fig. 2.6 in Tomkins (2024)). Observations are only used in the analysis if the station has reported snow for at least 4 hours to ensure we are using observations from consistent snow observations, and not any short-lived, low-impact events.

Following the guidelines of the Society of Automotive Engineers International Ground Deicing Committee and NCAR, we use a liquid equivalent precipitation rate threshold of $2.5 \mathrm{~mm~hr}^{-1}$ to distinguish heavy snow from light ($<1 \mathrm{~mm~hr}^{-1}$) and moderate ($1$-$2.5 \mathrm{~mm~hr}^{-1}$) snow (Rasmussen et al., 2001). We will use the term *heavy* to describe the snow rates $>2.5 \mathrm{~mm~hr}^{-1}$ liquid equivalent.

An example time series of liquid equivalent precipitation rates from Boston, MA on 7 February 2021 is shown in Fig.

3. Hourly observations that are used in the analysis are represented by a blue bar spanning the hour that the precipitation accumulates. Trace values ($0 \mathrm{~mm~hr}^{-1}$) are considered in the analysis (17 UTC), while observations when no precipitation fell are not (06-16 UTC). The example also shows observations not used in the analysis where the wind speed threshold is exceeded (22 UTC - 0 UTC) (plus sign in Fig. 3).

### 2.1.2 Regional radar mosaics

We use radar observations from the National Weather Service (NWS) Next-Generation Radar (NEXRAD) network in the northeast US to analyze features of the radar reflectivity field in winter storms. All NEXRAD data were obtained from the NOAA archive on Amazon Web Services (Ansari et al., 2018). The NEXRAD radars scan 360° in azimuth over sets of elevation angles that vary with the selected volume coverage pattern. In this analysis we use the elevation angle at 0.5° above the horizon as 0.5° is the lowest angle common to the different volume coverage patterns. We interpolate the single elevation in

the horizontal only. We found that interpolation among elevation angles, as is done in the NWS 3D interpolated radar reflectivity product (Smith et al., 2016), often smoothed out gradients bounding locally enhanced reflectivity features of interest.

Our radar data processing utilizes the open-source Python Atmospheric Radiation Measurement (ARM) Radar Toolkit (Py-ART) developed by the Department of Energy ARM Climate Research Facility (Helmus and Collis, 2016). We combine data from multiple radars in the northeast US into a mosaic on a Cartesian grid over an area of 1202 km x 1202 km with 2 km

horizontal grid spacing. The radars used to create the mosaics and an example mosaic are shown in Fig. 2. Regional mosaics





are generated for 5-10 min time steps during each storm. Tomkins et al. (2022) details the quality control, interpolation, and other data processing steps to produce the mosaics.

### 2.1.3 Objective detection of snow bands

Identifying and locating regions of locally-enhanced reflectivity can often be subjective and inconsistent from observer to ob-
server. To mitigate this, we developed a technique that objectively identifies local enhancements in radar observations (Tomkins et al., 2024). Previous methods to detect enhancements in enhanced reflectivity features in the rain layers of storms, such as convective precipitation cells, did not work well for detecting snow bands. Locally-enhanced reflectivity features in snow have more diffuse edges and smaller relative differences compared to the background values than convective cells in rain layers. The technique we employ uses two adaptive thresholds to identify objects based on their distinctness (faint or strong) from the
background average. We identify objects in a snow rate field that has been rescaled from reflectivity in order to define local enhancements based on spatial gradients that are roughly linear in snow rate. The algorithm outputs a feature detection field which classifies each grid point with radar echo as either part of a "faint feature", a "strong feature", or "background". Details on the feature detection algorithm including multiple examples of reflectivity fields, snow rate fields, and the identified features are provided in Tomkins et al. (2024). Example reflectivity, snow rate and feature classification fields are shown in Fig. 4.

Transitions between snow, rain, and partially-melted snow are common in northeast US winter storms which complicates the interpretation of reflectivity since melting and mixed precipitation have higher reflectivities than volumes of only ice or only liquid particles of the same mass. We identify regions of mixed or melting precipitation by combining information from the radar reflectivity and correlation coefficient dual polarization fields (Tomkins et al., 2022). We remove regions identified as mixed and melting precipitation from our analysis and reduce the visual prominence of these regions in our visualizations
using "image muting" (Tomkins et al., 2022).

### 2.2 Area × Time fraction

A main goal of this project is to understand the impact of locally-enhanced reflectivity features on the hourly surface snow rates. To accomplish this, we focus on the echo classified as features (from the methods discussed in Sec. 2.1.3) within 25 km of the ASOS station. The regional radar mosaics occur every 5-10 minutes, so in order to distill the echo area over the hour, we
computed an integrated metric called the area × time fraction. Figure 5 illustrates the echo area data that goes into calculating the area × time fraction. We examine separately the *feature area × time fraction* calculated for the echo classified as features (both strong and faint) and the *all echo area × time fraction* (strong and faint features plus background echo).

To calculate the area × time fraction we sum the area surrounding the station within 25 km radius for that hour and then we divide by the total seconds in an hour and the total area surrounding the station within 25 km which yields a unitless fraction.
Fig. 5c demonstrates 3 scenarios (of many) that would yield an area × time fraction of 0.5. A feature area × time fraction of 1 would indicate the entire surrounding region is filled with feature echo for the entire hour and a fraction of 0 would indicate no feature echo in the surrounding region for the entire hour.



## 2.3 Low pressure center tracks

For each winter storm, we computed the low-pressure center storm track from the ERA5 hourly reanalysis mean sea level
pressure (Hersbach et al., 2020) using the methods of Crawford et al. (2021). We subset the global 0.25° resolution ERA5
mean sea level pressure (MSLP) field to the region of Eastern North America (-90 to -60°E, 25 to 55°N). For a given winter
storm day, we track over ± 1 day to capture the full evolution of a given low pressure system. To find a single minimum
for a given time, we first find all the local minima in the MSLP field using a 200 km search radius. Unlike Crawford et al.
(2021), we are using the ERA5 data in native coordinates, so the grid size in km varies slightly between grid boxes. To avoid
finding pressure minima that skip along edges of our our defined Eastern North America region, we only consider minima
where at least 75% the 200 km search radius is within the region. Following Crawford et al. (2021), minima found from the
MSLP field are only considered if they have a pressure gradient of at least 7.5 hPa/1000 km. We compared the low pressure
center minimum to the set of pressures at 200 km range from the low pressure center. To meet the threshold criteria, the mean
difference between minimum pressure and the set of pressures at 200 km range must be at least 1.5 hPa. Once all the local
minima have been found for the entire period, we then loop through the period again to select a single minimum for each
time since there may be no minimum, one minimum, or multiple minima for each time. In the case of multiple minima we
choose the minimum that meets most of the following: minimum which is closest geographically to the previous minimum,
minimum with the strongest pressure gradient, and the minimum with the lowest MSLP value. If the closest, strongest, and
deepest minima are all different points, we choose the closest minima. If the next point is more than 200 km away from the
previous point, we consider this a new system and new track. We filter the tracks to remove tracks that travel less than 250 km
or exist for less than 6 hours. Finally, we remove any points on the tracks where the MSLP is greater than 1010 hPa.

All tracks between 2012 and 2023 that meet our winter storm criteria defined in Section 2.1 are shown in Fig. 6. The density
of tracks shows a clear tendency for low pressure centers to be near the coastline or over the ocean and to move southwest to
northeast roughly parallel to the coast (Fig. 6). This pattern is consistent with climatological studies of cyclone tracks (e.g.,
Bentley et al., 2019). There is a lack of low pressure locations along/to the east of the Appalachian mountains which we suspect
is due to our criteria for defining a snow storm for this study (Sec. 2.1). Since we are only looking at events where stations
along the coast and slightly inland in the northeast US produce snow, tracks that come up the Appalachian corridor may not
produce enough snow within our region of interest to be considered in our dataset.

Hourly liquid-equivalent snow rates from ASOS stations are plotted relative to the low pressure centers in Fig. 7. Most of
the winter season storms with appreciable snow had low center tracks offshore of the northeast US (Fig. 6). This tendency for
the low tracks to be offshore and the shape of the coastline yields a geographic bias of the sample to favor observations in the
northwest quadrant relative to the low pressure center (Fig. 7a). There is sufficient density of observations in the northwest
quadrant to suggest a gradient in snow rates with higher values more common closer to the low pressure center. The northeast
quadrant, where frontogenesis typically occurs associated with the warm front, had the highest median and mean hourly snow
rates among the four quadrants (see Table 6.1 in Tomkins (2024)). The low-centric spatial distributions of liquid equivalent
snow rate align with our expectations given typical spatial patterns of large-scale lifting associated with different storm stages





(Fig. 7). More active riming closer to the low, as found in Colle et al. (2014), would be consistent with heavier liquid water equivalents all other factors being equal but ASOS data does not let us assess this hypothesis.

## 2.4 NASA IMPACTS Airborne Radar Data

Vertical cross sections collected from radars aboard the NASA ER-2 aircraft deployed during the NASA IMPACTS field campaign (McMurdie et al., 2022) illustrate typical vertical structures and horizontal wind profiles observed in winter storms. There were 3 radars (4 frequencies total) that sampled storms during the campaign; EXRAD, HIWRAP (2 wavelengths), and CRS (see Table 2 for details). The examples we present here have a reflectivity panel from the radar with the longest wavelength that was available (usually EXRAD, and HIWRAP Ku-band if EXRAD was not available) as well as velocity and spectrum
width from the shortest wavelength that was available (usually CRS, and HIWRAP Ka-band if CRS was not available). Because the sensitivity is different for each radar (see Table 2) there is echo that is present in the velocity and spectrum width field that isn't present in the reflectivity field. When available, Velocity Azimuth Display (VAD) horizontal winds derived from the EXRAD scanning beam (Helms et al., 2020) are shown on the reflectivity cross sections and summarized with Contoured Frequency by Altitude Diagrams (CFADs; Yuter and Houze, 1995) in terms joint distributions of wind speed and height and
wind direction and height. The latter are shown in polar coordinates.

The ER-2 ground speed varies depending on wind speed and direction at flight level (i.e. slower in head winds and faster in tail winds). On average this value is $\sim 200 \mathrm{~m~s^{-1}}$. Actual ground speed values along each flight leg are converted to distance for plotting purposes. Section 3.2.2 shows 5 example transects during the NASA IMPACTS project. An additional 57 ER-2 transects with snow at the surface can be found in the supplement.

The velocity and spectrum width fields are shown to provide additional context for the examples. The velocity field can be used to infer regions of rain versus snow and Doppler spectrum width is a proxy for turbulence (Rauber and Nesbitt, 2018). To correct the spectrum width field to yield high quality data, we use equation 7 in Heymsfield et al. (1996) to isolate the spectrum width of the hydrometeors and remove the spectrum width broadening caused by the aircraft speed. In some cases, this correction causes the spectrum width numeric value to become $\leq 0 \mathrm{~m~s^{-1}}$ (i.e. a nonphysical value; see blank echo in Fig.
15c) which is ignored in the plotting as it represents signal below the noise floor.

## 2.5 Stony Brook Ka-band radar

Rapid update range-height indicator (RHI) scans from the Ka-band (35 GHz) scanning fully polarimetric radar (KASPR) located at Stonybrook University further illustrate the evolution of enhanced features in a winter storm (Figs. 18 and 19). The data from the ground-based KASPR radar has finer vertical and horizontal spatial resolution than the ER-2 airborne radar data
(Table 2).

The KASPR RHIs scan up and over the radar such that the azimuth angle of the radar beams within the RHI varies from 0-5° elevation angle in the southern part of the domain up to vertically-pointing right over the radar and then back down to 5-10° elevation angle at the northern horizon in about 30 seconds.



When the radar beam is vertical or nearly vertical, the measured values of Doppler velocity and spectral width combine vertical air motions with precipitation particle fall speeds. When the radar beam is closer to horizontal, the measured motions are more indicative of the horizontal wind.

The effect of the changing component of the wind that is sampled as the radar beam elevation angle changes is particularly noticeable in the RHIs of Doppler velocity data (Fig. 18b and 19b). Plotted velocity values in a given layer tend to be strongest at the left and right edges of the RHI where the beam is more horizontal and peter out as the beam becomes more vertical near the center of the RHI plot. Whereas in the ER-2 radar vertically-pointing data the Doppler velocity values were plotted with a range from -5 to 5 m s$^{-1}$, in these RHI plots, which are dominated by strong horizontal winds, the range plotted is -45 to 45 m s$^{-1}$. In the RHI plots, layers with low values of Doppler velocity indicate that the horizontal wind direction is close to perpendicular to the beam (i.e. in or out of the plane of the cross-section). A similar but less dramatic impact of beam angle on the measurements is apparent in the RHI spectral width plots. Although turbulence is usually close to isotropic, the combination of the air motion velocity spread with the strong signal from the often narrower precipitation fall speed spread tends to reduce the net spectral width magnitudes when the beam is pointed nearly vertical.

## 3 Results

The following results show that locally-enhanced reflectivity features are not consistently related to heavy surface precipitation rates. Most of the time, precipitation rates are low, even when locally-enhanced reflectivity features are present. Vertical cross sections from airborne and ground-based radars illustrate that the local enhancements in reflectivity within snow are tilted and smeared on the way to the surface. The lack of vertical continuity of reflectivity enhancements helps to explain the lack of relationship between local enhancements observed by radar and heavier surface snow rates.

### 3.1 Enhanced reflectivity features in horizontal maps and their relation to surface snowfall rates

Figure 8a shows the 2D distribution of all features (faint + strong) area × time fraction versus liquid equivalent precipitation rate. The quadrants in Fig. 8a are summarized in a bar chart in Fig. 8c. Considering the previous literature focused on snow banding in winter storms, we were expecting to see a trend in the joint distribution indicating a relationship between increasing feature area × time fraction and increasing precipitation rate, however this is not the case for the large sample size examined in this study. Most points (89.1%) are clustered in the lower left box where both area × time fraction and precipitation rate are low. There are some observations where the area × time fraction and precipitation rate are high, however they only account for ∼1.5% of the observations when the feature area × time fraction is > 0.

*Heavy hourly snow rates in northeast US winter storms are rare*. Over all the hourly observations presented here, heavy snow rates (> 2.5 mm hr$^{-1}$) occurred < 4% of the time. This analysis indicates that anecdotal evidence from case studies in the literature showing a strong relation between heavier snow at the surface and enhanced reflectivity features aloft is not representative for a large sample size of 7606 hours, 264 storms, obtained over 11 years. Figure 8a,c indicates that *equating snow bands with heavy snow will usually lead to over prediction of hourly snowfall rates*. Our large sample size shows that 3



out of 4 times situations with feature area $\times$ time fractions $> 0.5$ have liquid equivalent snow rates $< 2.5$ mm hr$^{-1}$. More hours with heavy snow at the surface occurred associated with smaller feature areas (364 hours) as compared to larger feature areas (113 hours).

Since locally-enhanced reflectivity features are not particularly helpful at identifying regions of heavy snow rates, we calcu-
late the area $\times$ time fraction for *all echo areas* to see if there are patterns present (Fig. 8b,d). The patterns in this 2D distribution indicate that heavy snow rates ($> 2.5$ mm hr$^{-1}$) are more common when there is a large echo area $\times$ time fraction (i.e. there is a lot of echo surrounding the station for a longer duration). This suggests that it is more useful to focus on the duration of all echo over a location rather than the size, duration, and location of just the locally-enhanced reflectivity features for predicting where higher snowfall accumulations may occur. The distribution of snowfall rates has a strong skewness to low values, 75%
of the time it is snowing at a rate no more than 1 mm hr$^{-1}$. So even if there is enhanced reflectivity feature area it is more likely associated with a low snow rate than a heavy snow rate.

To provide context for the 2D distribution, we present representative examples of each quadrant in Fig. 8a. The first is an example when high feature area $\times$ time fraction and heavy snowfall are observed over an hour (Fig. 9). Panel a in Fig. 9 illustrates a large, strong feature over the Albany, NY ASOS station which contributes to a feature area $\times$ time fraction of 0.62
over the hour and coincides with an hourly liquid equivalent snowfall rate of 3.6 mm hr$^{-1}$. Later in this event, there are several hours at this station that were not included in our analysis because the wind speed was too high (blue plus signs in Fig. 9c). The scenario of high feature area, heavy snowfall occurs in only 1.5% of hours in our dataset. In this specific example, the feature resembles a primary band and is likely forced by frontogenesis which we would expect to fall into this category.

The second example is from an event on 26 January 2021 where several strong and faint features are moving through the region (Fig. 10a). In the snapshot in Fig. 10a, there is a strong feature over the Providence, RI ASOS station which contributes to a feature area $\times$ time of 0.58 over the hour (Fig. 10). While the feature area $\times$ time fraction is high ($> 0.5$) in this example, the snow rate is 2 mm hr$^{-1}$ (Fig. 10c). Scenarios with high feature area $\times$ time fraction and a low/moderate snow rate represent 4.6% of hours in our dataset.

The next example from 2 February 2021 represents the most common occurrence in our dataset where a low feature area $\times$ time fraction and a low/moderate snow rate is observed over the hour. This example from Lebanon, NH shows only a small faint object in the vicinity of the ASOS station and has a feature area $\times$ time fraction of 0.2 over the hour and a snow rate of 1.3 mm hr$^{-1}$ (Fig. 11). For the remainder of the event there are no features in the vicinity of the ASOS station and the snow rate remains low/moderate (Fig. 11c). Scenarios when the feature area $\times$ time fraction is low and the snow rate is low account for 89% of observations.

Lastly, we have an example when the feature area $\times$ time fraction is low but the snow rate is heavy which occurs in 4.8% of our observations. This example from Worcester, MA shows an hour where the liquid equivalent precipitation rate was 9.7 mm hr$^{-1}$ but the feature area $\times$ time fraction was only 0.12 (Fig. 12). There are a few faint features in the area, but overall the echo is patchy. The all echo area $\times$ time fraction was 0.9. The time series shows that there are several times in this event where the snow rate was heavy but there were little or no features present over the station. Scenarios with small
feature area $\times$ time fraction are commonly have a long duration of background echo rather than mostly locally-enhanced





reflectivity features. It is also possible that in this case that the snow storm is shallow and the radar beam height is above any locally-enhanced features.

### 3.1.1 Sensitivity Tests

We tested the sensitivity of the results in Fig. 8 by varying various thresholds used in the calculations and these adjustments
did not appreciably change our results (Figs. 5.10-5.14 in Tomkins, 2024). We varied the radius around the ASOS station to 12.5, 25, and 50 km to account for the horizontal advection of snow, the time lag between the radar and ASOS observations (0-, 1-, and 2-hour), and the number of hours of precipitation accumulation (1-, 2- and 3- hours). We also subset by radar beam altitude above the ASOS station, and percent of mixed precipitation echo removed over the hour. For example, applying thresholds based on the average beam height of the radar echo changed the fraction of hours in the high area × time fraction-
heavy snow rate category to 1.8% for $\leq 1000$ m, 1.7% for $\leq 2000$ m, and 1.5% for $\leq 3000$ m. While these sensitivity tests yielded slightly different numbers for each quadrant in the 2D distributions of feature area × time fraction versus liquid water equivalent precipitation rate, the key findings from Fig. 8 are robust relative to the tested variations.

## 3.2 Vertical structures in reflectivity are tilted and smeared

The lack of relationship between ground-based scanning radar observed locally-enhanced reflectivity features and snow rates
indicates that the radar observations above the surface are not necessarily consistent with snow rates observed directly below at the surface. There are several factors that complicate the relationship between reflectivity and snowfall rate that are not present in the relationship between reflectivity and rain rate. Changes in radar reflectivity do not necessarily translate to changes in ice mass. As is discussed in Sec. 1, processes such as aggregation can increase the reflectivity without increasing the mass per unit volume (Table 1). The typical fall speeds of raindrops ($\sim$2-8 m s$^{-1}$, depending on raindrop size) often yield vertical column
continuity of enhanced reflectivity features in rain. In contrast, the slower fall speeds of snow ($\sim 1 \pm 0.5$ m s$^{-1}$) commonly yield sufficient time for horizontal advection of the snow form curved ice streamers. Falling snow particles can be blown sideways more than 50 km horizontally from the locations where they first achieve precipitation size near the top of the storm.

Locally enhanced reflectivity features in snow tend to be tilted and smeared by the wind shear (changes in the wind speed and direction with height) between echo top and the surface. High spatial resolution vertical cross-sections of radar data from winter
storms indicate that snow rarely falls straight down to the surface. In the following two sections, we feature examples of high vertical resolution cross-sections within snow storms that illustrate the lack of vertical column continuity in locally-enhanced reflectivity.

### 3.2.1 Vertical structures observed by airborne radar data

As aircraft equipped with vertical-pointing radar fly along, they obtain a slice through the atmosphere. For the NASA ER-2
which has an ground speed of $\sim 200$ m s$^{-1}$, each 50 km in distance along the leg represents 5.6 minutes of flight time. Adjacent portions along the track can be considered to be within a few minutes coincident in time.



The first example from 5 February 2020 in the Midwest illustrates several types of variations in the structures in reflectivity field (Fig. 13). At the beginning of the transect, when the aircraft is traveling from the edge of the echo in the regional map (Northern Illinois; Fig. 13d) the vertical cross sections indicate that the echo aloft is not reaching the surface (0-40 km, 0-1 km altitude in Fig. 13a). At ∼90 km along flight track, there are regions where there is echo reaching the surface but "holes" in the echo aloft. Other regions along the cross section show consistent echo through the vertical column and indicate lots of wind shear illustrated by the wind barbs and bends in the local variations of the reflectivity field itself (i.e. 125-175 km in Fig. 13a). It is important to note that the cross sections are plotted in a 3:1 aspect ratio so vertical features that are tilted are even more tilted in reality (see triangle icons next to Fig. 13c for visualization). CFADs of the wind speed (Fig. 13e) and wind direction (Fig. 13f) along the track indicate considerable vertical wind shear. Horizontal wind speeds reach around 30 m s$^{-1}$ at 5 km altitude near the top of the echo. The wind direction at 5 km altitude is roughly perpendicular to the direction the aircraft is traveling indicating that snow particles forming aloft will be transported out of the page by the wind as they descend to the surface (Fig. 13b, c).

The next example from 7 February 2020 shows a transect over New York (Fig. 14). In this example, the reflectivity is more uniform compared to the previous example (Fig. 14a). The aircraft is traveling westward from a region of higher reflectivity in central New York to a region of weaker reflectivity in western New York (Fig. 14a, d). This case was examined in Colle et al. (2023) for the lack of snow banding despite considerable frontogenesis present (values > 10 K (100 km)$^{-1}$ (3 hr)$^{-1}$). The CFADs of the wind speed and direction from the transect indicate speed shear and some directional shear (Fig. 14e,f). Similar to Fig. 13, the wind direction is roughly perpendicular to the direction the aircraft is flying, indicating that the particles forming aloft are advected away from the flight path – and hence the vertical plane of the cross-section – as they fall (Fig. 14).

An example from 23 January 2023 shows a case where the aircraft flew parallel to strong, banded features in Maine and New Hampshire (Fig. 15e). The reflectivity cross section shows ice streamers that are tilted as they approach the surface. The top of the ice streamers are likely generating cells associated with overturning circulations near cloud top (Fig. 15a). The velocity and spectrum width cross sections indicate some overturning circulations near echo top between 150-200 km along the flight track (Fig. 15b,c). A non-uniform beam filling (NUBF) correction was applied to the velocity data and the circulation features show up in all wavelengths indicating that these velocity features are not NUBF (G. Heymsfield and M. Walker McLinden, personal communication, July 2023). Additionally, there are local enhancements in velocity,tilted blue linear features in b), and corresponding higher spectral width in c) within the tilted ice streamers as particles originating in the generating cells descend to the surface. In this example, the wind direction is opposite to the direction the aircraft is flying near the top of the echo and roughly in the same direction as the aircraft towards the surface. Unlike the previous two examples, this implies that the particles are advected along the same plane of the cross section as they fall.

The remaining examples do not have VAD horizontal wind data available. We include a panel of the feature detection field instead for context. Figure 16 shows an example from a 300 km long flight track that flew over some faint enhanced reflectivity features in the Gulf of Maine. The regional radar reflectivity indicates that this storm had weaker and more patchy echo compared to the previous examples (Fig. 16d). The radar cross sections show shallower echo tops (∼5 km altitude) compared





to the previous examples. Between 125 km and 175 km along flight track, enhanced reflectivity features tilt to the right and then to the left. There are weak echo holes throughout the cross-section.

Figure 17 shows an example from 17 February 2022 when the aircraft sampled a faint, banded feature over Lake Michigan (200-215 km along flight track). The flight track began over central Indiana (southeast) over surface rainfall which transitioned to surface snow at ∼48 km along the flight track. This surface rain portion of the flight track is depicted as an image muted area in gray in Fig. 17d and has a radar bright band, and high Doppler velocities and spectral width in the rain layer between 0-48 km (Fig. 17abc). Along the flight track within the snow layer between 6 and 2.5 km altitude, locally enhanced reflectivity features in snow tend to tilt to the right (towards northwest). At the northwest end of the flight track when the aircraft approaches the faint enhanced feature in the map (Fig. 17e), there is a layer of locally-enhanced reflectivity at ∼2.5 km altitude between 150-250 km (Fig. 17a) with weaker to no echo below it. It is possible that the faint feature in the map manifests as the scanning radar intersects part of this elevated region of enhanced reflectivity.

As part of this study, we examined all transects from the NASA IMPACTS campaign which included snow reaching the surface. In addition to Figs. 13-17 there are 53 additional transects which are shown in the supplementary material. IMPACTS flight legs that pass over regions with just rain at the surface are not included. For each transect in the supplement, we present the reflectivity, velocity, spectrum width, and LDR fields from the instrument with the shortest wavelength available (usually CRS, but occasionally HIWRAP Ku-band) and the reflectivity field from the instrument with the longest wavelength available (usually EXRAD, but occasionally HIWRAP Ka-band). We include snapshots of the NEXRAD regional maps to provide context for the transects. If available, we include the VAD winds on the EXRAD reflectivity transects and CFADs of wind speed and direction to summarize. Each transect in the supplement is annotated with a vertical, dashed, black line which annotates where the enhanced reflectivity features are in the regional maps. Full details are available in the supplementary material. The additional examples shown in the Supplement are consistent with the findings illustrated in Figs. 13-17.

### 3.2.2 Sequences of vertical structures observed from ground-based radar data

Fast update RHIs from ground-based radar can obtain individual vertical cross-sections that complete in 10s of seconds and yield sequences of RHIs that can address storm evolution along the vertical plane at time scales < 1 minute. Within snow storms noticeable structural changes along a given vertical cross-section from a combination of storm advection and evolution usually have time scales of several min or more.

We show examples of fast update RHIs from the Stony Brook University KASPR from a storm that spanned 40 hours on 31 January to 2 February 2021. An animation of the sequence of RHIs for the full storm period, as well as two 2-hour periods corresponding to Figs. 18 and 19 is included in the Video Supplement. While RHIs have often been obtained by research radars in winter storms, this is one of the few examples with a fast enough update to discern how a individual features change with time along the plane of the RHI vertical cross-section.

During the 31 January to 2 February 2021 storm, the KASPR scan strategy consisted of the following sequence that was repeated every 15 minutes: PPI at 15 ° elevation angle (50 seconds), set of 8 cross-band RHIs (6 minutes), vertically pointing mode (2 minutes), set of 8 cross-band RHIs (6 minutes). KASPR's mechanical antenna took a few seconds to reverse direction





at the end of each RHI sweep. Each single RHI sweep took ∼30 seconds (scan rate of 4-6 sec/deg) and can be considered to be similar to a snapshot.

Below the cloud-top generating cells in first example at 1558 UTC 1 February 2021 (Fig. 18), the reflectivity enhancements bend with the wind patterns which switch direction several times through the depth of the echo. The spectrum width is enhanced at echo top associated with overturning circulations within the generating cells (Fig. 18a). Other narrow layers of enhanced
spectrum width are present at several altitudes including close to the surface (Fig. 18a). The Doppler velocity field helps to illustrate the horizontal winds and how they vary with height (Fig. 18b). Near cloud top the winds are left to right, corresponding to a wind with a southerly component (wind coming from the south). In the layer between 2.3 and 3 km AGL the magnitude of the velocity is near zero which indicates the wind direction is pointing perpendicular to the radar beam. Between the surface and 1 km altitude there is shift in the wind direction to a wind with a North component illustrated by the change in sign of
the Doppler velocity. The reflectivity field shows a lot of detail, including ice streamers that manifest from the overturning circulations near echo top (Fig. 18c). The ice streamer features become tilted and smeared on the way to the surface, likely due to the distinct layers in the wind profile illustrated in the Doppler velocity field. In this example, the cross sections are plotted in a 1:1 aspect ratio so the tilt of the features is shown as it occurs in reality. As the snow particles within the enhanced reflectivity features descend in the storm they move much faster sideways than they do vertically. Unless the horizontal winds
are very weak, snow cannot fall "straight" down in a column. It is also important to note that the precipitation particles are not all falling within the plane of the RHI. In particular, reflectivity values at altitudes with layers of near zero Doppler velocity are potentially either moving in or out perpendicular to the cross-section. Whenever the wind direction changes between layers, the trajectories of individual precipitation particles turn with the wind, yielding complex 3D trajectories between their origin near cloud top and the surface.

In the second example a few hours later at 1928 UTC (Fig. 19), the cloud-top generating cells are still evident and the bending of the enhancements to the left then the right is more evident. The echo is slightly shallower overall and the reflectivity field is more variable near cloud top varying from close to minimum detectable echo at -5 dBZ to close to 25 dBZ (Fig. 19c). There is some locally higher spectral width and likely turbulence close to 5 km altitude near echo top at the location of generating cells near -24 to -26 km and directly above the radar near 0 km (Fig. 19a). There is a discontinuity in the direction
of tilt of ice streamers within the reflectivity field at about 3 km altitude from toward the right (North) to toward the left (South) corresponding to the wind shift observed in the Doppler velocity field (Fig. 19b). Similar to at 1558 UTC, the wind also sharply shifts direction to a North wind at about 1 km altitude above the surface.

The aircraft radar cross sections from the NASA IMPACTS campaign and RHI scans from the KASPR radar illustrate how features in the snow reflectivity field are tilted and smeared in winter storms. The lack of vertical column continuity of enhanced
reflectivity aloft to the surface complicates the interpretation of reflectivity from PPI scanning radar in snow and provides a physical explanation for why we are not seeing a strong relationship between enhanced reflectivity aloft and snow rates at the surface.



## 4 Conclusions

Where and when heavy snow is likely to occur in northeast US winter storms is both a high impact and thorny problem. We
analyzed observations of a large sample of snow storm events with negligible orographic forcing in the northeast US to improve
the understanding of how hourly snowfall rates relate to structural characteristics of these winter storms. Data are from 264
storm days over 11 years (2012-2023) which yielded over 7500 time-spatial matched pairs of hourly surface observations and
radar data that included enhanced reflectivity features in snow. Motivated by previous work, we examined mesoscale snow
bands (locally-enhanced reflectivity features) - a "prime suspect" in the occurrence of heavy snowfall. Most of the time in
northeast US winter storms, the snow rates are low (75% of hours had liquid equivalent snow rates less than 1 mm hr$^{-1}$
[horizontal annotations on light blue bars in Fig. 8c, d]) and only 4% of hours had heavy snow rates > 2.5 mm hr$^{-1}$ (Fig.
8). Our data set excludes conditions with high surface winds (> 5 m s$^{-1}$) so these data are representative of non-blizzard
conditions. Consistent with previous work, higher snow rates are more likely to occur closer to the low-pressure center (within
250-500 km), especially in the northwest and northeast quadrants of the storm.

Regional radar mosaics over the northeast US were created from the NWS NEXRAD network that included output from
two new objective image processing methods developed specifically for this work: the removal of mixed precipitation ar-
eas (Tomkins et al., 2022), and the objective identification of locally-enhanced reflectivity features in snow (Tomkins et al.,
2024). Potential associations between radar reflectivity structures in the vicinity of ASOS stations and hourly snow rates were
examined in terms of incidence of locally-enhanced reflectivity features as well as all snow echo.

Evidence from our analysis demonstrates that operational PPI scanning radar observed locally-enhanced reflectivity features
within these snow storms are not consistently associated with heavy snowfall rates at the surface. When enhanced reflectivity
features pass over over a location (area × time fractions > 0.5), only 1 out of 4 occurrences have hourly heavy surface snow
liquid water equivalents ($\geq$ 2.5 mm hr$^{-1}$) (Fig. 8a, c). Evidence from fine spatial resolution vertical cross-sections obtained in
57 airborne radar transects by the NASA ER-2 aircraft during the IMPACTS field campaign, and from ground-based KASPR
scanning fast update RHIs shows why the association between snow bands and surface snow rates is weak. Simply put, snow
particles rarely fall straight down (Figs. 13-19). Locally enhanced reflectivity features in snow are typically *tilted and smeared*
as the precipitation-size ice is blown sideways by the horizontal wind for 10s of km. Precipitation-size ice has typical fall
speeds of 1 m s$^{-1}$ ± 0.5 m s$^{-1}$ (Fitch et al., 2021). It takes an ice particle 16.6 minutes to descend 1 km if it is falling at
1 m s$^{-1}$. In that time, it will move laterally 10 km in a horizontal wind of 10 m s$^{-1}$. In comparison, a rain drop falling at
5 m s$^{-1}$ will only take 3.3 minutes to descend 1 km, and would move laterally only 2 km in a horizontal wind of 10 m s$^{-1}$.

Numerous previous studies of radar observations going back to the 1950's Marshall (1953) have shown that ice streamers
are tilted as they are blown side ways with the horizontal wind. The crucial nuance revealed by the analysis of large sample
sizes in our study is that the extent of the smearing during that process usually yields few situations where the locally higher
reflectivity aloft results in high snowfall rates at the surface. A set of snow particles in a volume at 2 km altitude are unlikely
to all arrive together at the surface because of variations in their fall speeds and small scale variations in the horizontal wind
including turbulence. An additional factor complicating interpretation of enhanced radar reflectivity in snow is that it does not





always imply increased ice mass (i.e. aggregation, Table 1). Our work suggests that it is more useful to focus on the duration of snow radar echo over a given location to predict higher snowfall accumulations rather than focus on the locally-enhanced reflectivity features.

While case studies can be informative, those that focus on storms with heavy snowfall may not be representative of a large sample of snow storms that include a range of storm intensities and durations. Primary bands are associated with strong frontogenesis and high snowfall accumulation but don't occur that often (Novak et al., 2004; Kenyon et al., 2020; Ganetis et al., 2018). Multi-bands occur more frequently but are found in environments with and without frontogenesis (Ganetis et al., 2018). Hence, in non-orographic storms, the presence of snow bands in radar reflectivity cannot be used as a proxy for sustained

upward motions.

When students learn how to interpret radar reflectivity, the vast majority of examples used in training are from rain layers. The close association between locations with stronger reflectivities several km above the surface and higher surface precipitation rates while appropriate for rain is not for snow. Practitioners and automated systems interpreting observed radar data would benefit from utilizing different "rules of thumb" for rain versus snow.

*Code and data availability.* Data: The NWS NEXRAD Level-II data used in Figs. 2, 4, 5, 13, 14, 15, 16, and 17 can be accessed from the National Centers for Environmental Information (NCEI) at https://www.ncei.noaa.gov/products/radar/next-generation-weather-radar. The radar composites created from the NEXRAD Level-II data used in Figs. 2, 4, 5, 13, 14, 15, 16, and 17 can be accessed from a Dryad repository at https://doi.org/10.5061/dryad.rbnzs7hj9. The NWS ASOS data used in Figs. 2.1.1, 7, 5, 9, 10, 11, 12 can be accessed from NCEI at https://www.ncei.noaa.gov/products/land-based-station/automated-surface-weather-observing-systems. The ERA5 data used to cre-

ate the low pressure tracks in Fig. 6 can be accessed from the Copernicus Climate Data Store at https://cds.climate.copernicus.eu/datasets/reanalysis-era5-single-levels?tab=overview. The derived low pressure tracks created from the ERA5 data can be accessed from an OSF directory at https://osf.io/az5w2/. The ER-2 radar data used in Figs. 13–17 can be accessed from NASA at https://ghrc.nsstc.nasa.gov/uso/ds_details/collections/impactsC.html.

Code: The functions used to create the feature detection fields are available in Py-ART (Helmus and Collis, 2016) and be accessed via

https://arm-doe.github.io/pyart/API/generated/pyart.retrieve.feature_detection.html. An example of how to use the function is provided here: https://arm-doe.github.io/pyart/examples/retrieve/plot_feature_detection.html.

*Video supplement.* List of animations with captions and filenames

All animations can be viewed at: https://doi.org/10.5446/s_1851. Individual animations can be viewed by following the DOI URL.

Animation-Figure-9: Animated plot of Fig. 9 showing an example when a heavy snow rate ($\geq$ 2.5 mm hr$^{-1}$) and high feature area $\times$ time fraction ($>$ 0.5) are observed over an hour. (Top panel) Feature detection field from NEXRAD regional mosaic animated over 16 December 20:00 UTC to 17 December 20:00 UTC with Albany, NY ASOS station (KALB) and 25 km radius annotated in purple and (bottom panel) time series of hourly precipitation rate between 16 December 20:00




UTC and 17 December 20:00 UTC from KALB (blue annotations) and area of each feature category within 25 km of KALB (yellow: strong area, orange: faint area, and teal: background area). In the bottom panel, purple vertical line indicates time of specific NEXRAD regional mosaic in the top panel.

Title: 17 December 2020 area × time fraction example DOI: https://doi.org/10.5446/69385

Animation-Figure-10: Animated plot of Fig. 10 showing an example when a low/moderate snow rate ($< 2.5$ mm hr$^{-1}$) and high feature area × time fraction ($> 0.5$) are observed over an hour. (Top panel) Feature detection field from NEXRAD regional mosaic at 26 animated over 26 January 12:00 UTC to 27 January 00:00 UTC with Providence, RI ASOS station (KPVD) and 25 km radius annotated in purple and (bottom panel) time series of hourly precipitation rate between 26 January 12:00 UTC and 27 January 00:00 UTC from KPVD (blue annotations) and area of each feature category within 25 km of KPVD (yellow: strong area, orange: faint area, and teal: background area). In the bottom panel, purple vertical line indicates time of specific NEXRAD regional mosaic in the top panel.

Title: 26 January 2021 area × time fraction example DOI: https://doi.org/10.5446/69386

Animation-Figure-11: Animated plot of Fig. 11 showing an example when a low/moderate snow rate ($< 2.5$ mm hr$^{-1}$) and low feature area × time fraction ($\leq 0.5$) are observed over an hour. (Top panel) Feature detection field from NEXRAD regional mosaic animated over 02 February 00:00 UTC to 03 February 00:00 UTC with Lebanon, NH ASOS station (KLEB) and 25 km radius annotated in purple and (bottom panel) time series of hourly precipitation rate between 02 February 00:00 UTC and 03 February 00:00 UTC from KLEB (blue annotations) and area of each feature category within 25 km of KLEB (yellow: strong area, orange: faint area, and teal: background area). In the bottom panel, purple vertical line indicates time of specific NEXRAD regional mosaic in the top panel.

Title: 02 February 2021 area × time fraction example DOI: https://doi.org/10.5446/69387

Animation-Figure-12: Animated plot of Fig. 12 showing an example when a heavy snow rate ($\geq 2.5$ mm hr$^{-1}$) and low feature area × time fraction ($\leq 0.5$) are observed over an hour. (Top panel) Feature detection field from NEXRAD regional mosaic animated over 27 January 00:00 UTC to 27 January 18:00 UTC with Worcester, MA ASOS station (KORH) and 25 km radius annotated in purple and (bottom panel) time series of hourly precipitation rate between 27 January 00:00 UTC and 27 January 18:00 UTC from KORH (blue annotations) and area of each feature category within 25 km of KORH (yellow: strong area, orange: faint area, and teal: background area). In the bottom panel, purple vertical line indicates time of specific NEXRAD regional mosaic in the top panel.

Title: 27 January 2021 area × time fraction example DOI: https://doi.org/10.5446/69388

Animation-Figure-13: An animation of a flight leg from 05 February 2020 between 21:11:29 and 21:31:09 UTC (the flight leg before the one shown in Fig. 13). (Left panel) Image muted reflectivity from NEXRAD radar mosaic, green line indicates path of ER-2 flight. Right panels are reflectivity fields from the radars on the ER-2, wavelength decreases from top to bottom. Vertical line in right panels indicates location of ER-2 path and small x marks the height of the radar beam in the NEXRAD mosaic.

Title: 05 February 2020 NEXRAD & EXRAD animation DOI: https://doi.org/10.5446/69554



Animation-Figure-18: Animated plot of Fig. 18 showing (a) Spectrum Width [m s$^{-1}$], (b) Doppler Velocity [m s$^{-1}$], and (c) Reflectivity [dBZ] RHIs from KaSPR radar at Stonybrook University between 15:00 and 17:00 UTC on 1 February 2021. RHIs are scanned up and over the radar (at 0 km on x-axis). Radar beam is partially blocked near edge of scan on right side. Plotted in a 1:1 aspect ratio.

Title: 01 February 2021 KaSPR RHI example 1 DOI: https://doi.org/10.5446/69389

Animation-Figure-19: Animated plot of Fig. 19 showing (a) Spectrum Width [m s$^{-1}$], (b) Doppler Velocity [m s$^{-1}$], and (c) Reflectivity [dBZ] RHIs from the KaSPR radar at Stonybrook University at between 18:30 and 20:30 UTC on 1 February 2021. The RHIs are scanned up and over the radar (at 0 km on x-axis). Radar beam is partially blocked near edge of scan on right side. Plotted in a 1:1 aspect ratio.

Title: 01 February 2021 KaSPR RHI example 2 DOI: https://doi.org/10.5446/69391

Animation-Figure-19-19-full: Animated plot of full event from Fig. 18 and 19 showing (a) Spectrum Width [m s$^{-1}$], (b) Doppler Velocity [m s$^{-1}$], and (c) Reflectivity [dBZ] RHIs from the KaSPR radar at Stonybrook University between 10:00 UTC on 31 January 2021 and 11:00 UTC on 2 February 2021. The RHIs are scanned up and over the radar (at 0 km on x-axis). Radar beam is partially blocked near edge of scan on right side. Plotted in a 1:1 aspect ratio.

Title: 01 February 2021 KaSPR RHI full example DOI: https://doi.org/10.5446/69390

*Author contributions.* LMT and SEY conceptualized the project and designed the methodology with input from MAM. LMT wrote the Python software with input from SEY and MAM. MO curated the KaSPR data and CNH curated the NASA IMPACTS EXRAD VAD data. LMT prepared the manuscript and the figures. All authors contributed to editing and review.

*Competing interests.* The authors declare that they have no conflict of interest.

*Acknowledgements.* The authors thank Kevin Burris and Luke Allen for their helpful feedback and support throughout the entirety of this project. Anya Aponte-Torres, Declan Crowe, Jordan Fritz, Cameron Gilbert, Wayne Johnson, Rebecca Moore, McKenzie Peters, and McKenzie Sevier assisted with data processing. Jordan Fritz and Rebecca Moore produced and compiled the ER-2 flight leg plots for the supplemental material. The authors also express their sincere appreciation to the entire the NASA IMPACTS project team. This paper greatly benefited from the data collected during 3 seasons of research aircraft deployments and the insightful conversations with science team members. This research has been supported by the National Science Foundation (AGS-1905736), the National Aeronautics and Space Administration (80NSSC19K0354), and the Center for Geospatial Analytics at North Carolina State University.



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





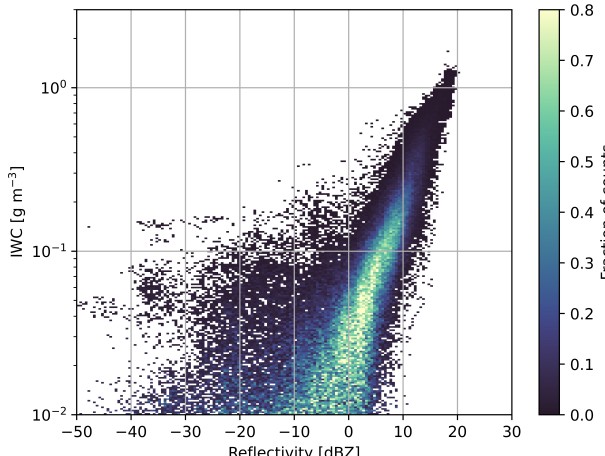

**Figure 1.** Joint distribution based on over 100,000 coincident aircraft observations of 1 Hz Nevzorov probe Ice Water Content samples and Wyoming Cloud Radar reflectivities obtained in light snow by the University of Wyoming King Air during the SNOWIE project in the mountains of western Idaho. Reflectivity values at the aircraft flight level were calculated by linear interpolation between the first valid range gates above and below the aircraft. Adapted from Figure 6 in Zaremba et al. (2023).



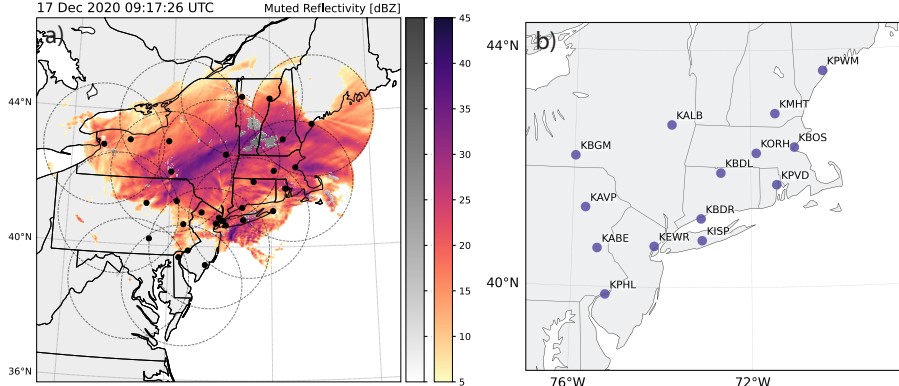

**Figure 2.** (a) An example of the reflectivity mosaic from 27 December 2020 09:17 UTC. Black dashed circles indicate the 200 km extent from each radar used in the mosaics. Black dots indicate ASOS stations where hourly snow rate is used in the analysis. (b) map of ASOS stations where daily snowfall accumulation is used to define a winter storm in this analysis.





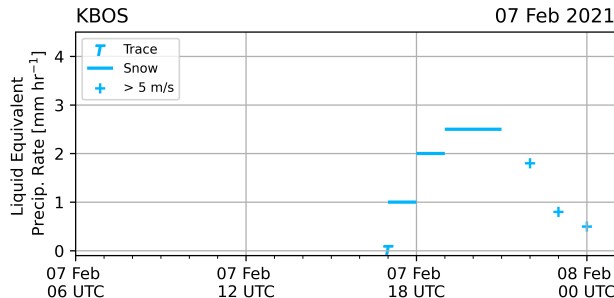

**Figure 3.** Example time series of liquid equivalent precipitation rate $[\mathrm{mm\,hr}^{-1}]$ from the Boston, MA ASOS station (KBOS) from 7 February 2021 06:00 UTC to 8 February 2021 01:00 UTC. Precipitation rates are represented as a bar spanning the hour they were accumulated. Plus signs indicates times when the wind speed is $> 5\ \mathrm{m\,s}^{-1}$ and $T$ indicates times when trace amounts of precipitation where observed.





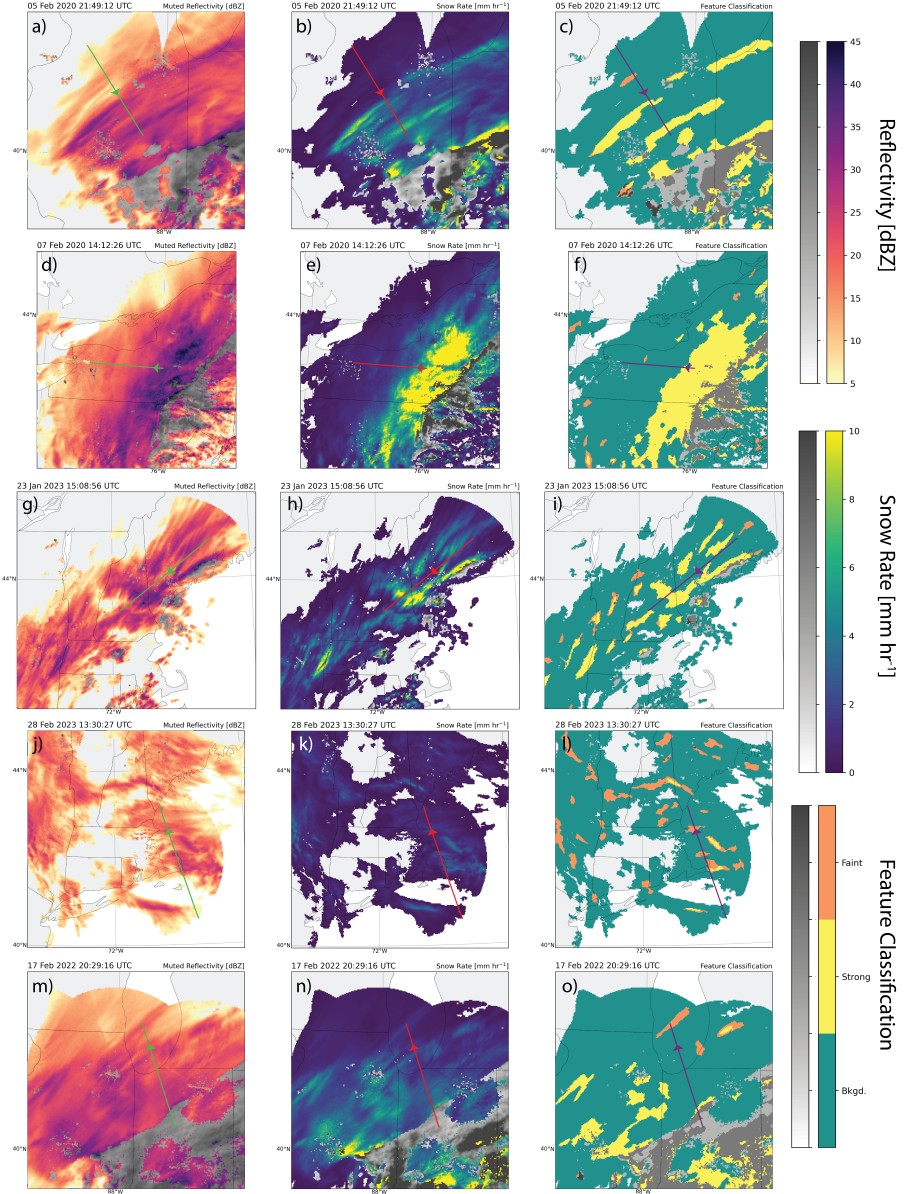

**Figure 4.** Locally-enhanced reflectivity feature classification for the examples presented in Sec. 3.2.2 (Figs. 13–17). Each row shows a different example. (a)-(c) 5 February 2020 21:49:12 UTC, (d)-(f) 7 February 2020 14:12:26 UTC, (g)-(i) 23 January 2023 15:08:56 UTC, (j)-(l) 28 February 2023 13:30:27 UTC, (m)-(o) 17 February 2022 20:29:16. Left column shows step 1, the input radar reflectivity (dBZ). Middle column shows step 2, the reflectivity field rescaled to estimated snow rate [mm hr$^{-1}$]. Right column shows step 3, the feature classification into faint and strong local features and background values. Each field is imaged muted to reduce the visual prominence of melting and mixed precipitation (gray scale) following Tomkins et al. (2022). Green, red, and purple lines indicate NASA ER-2 flight paths and arrow heads indicate location of the aircraft at the time of each plot. The feature classification method is detailed in Tomkins et al. (2024).





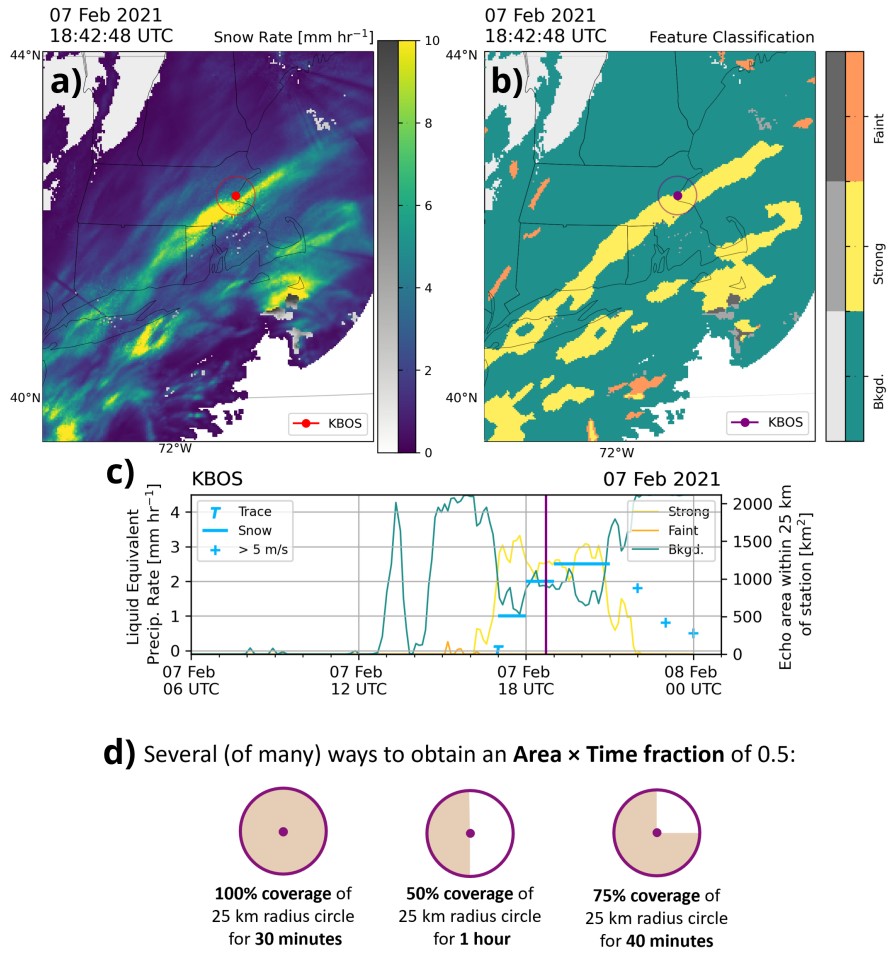

**Figure 5.** Illustration of how areal radar data and point measurements of snowfall rates are combined. (a) Shows the snow rate field and (b) shows the feature detection field centered on Massachusetts for 7 February 2021 18:42 UTC. (c) The corresponding time series of ASOS hourly precipitation rate valid from 7 February 2021 06 UTC to 08 February 2021 01 UTC and echo areas calculated within 25 km of Boston, MA ASOS station (KBOS: red dot in (a) and purple dot in (b)). Lines in time series (c) correspond to background echo area (teal), strong area (yellow) and faint area (orange) within 25 km of the KBOS station (rings in (a) and (b)). Purple vertical line annotated on time series (c) indicates the time of the maps. (d) Several examples of echo coverage over a given time that would yield an area × time fraction of 0.5.





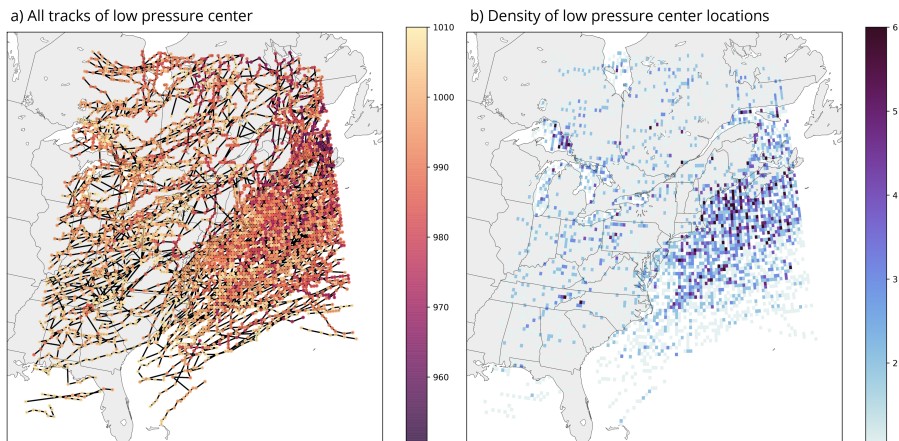

**Figure 6.** Map of (a) all tracks of low pressure centers and (b) density of low pressure center locations for events between 2012-2023 for the region shown. Color shading indicates low pressure magnitude in hPa units.

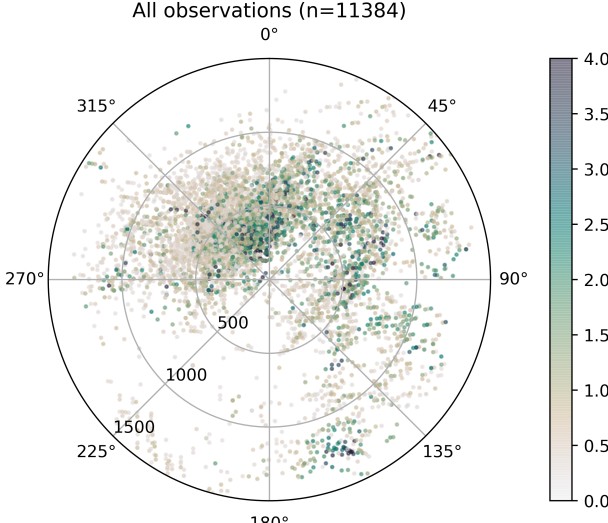

**Figure 7.** Lagrangian low-centric framework plot of hourly liquid equivalent snowfall rates in units of $m\,s^{-1}$. The plot's center represents the tracked low pressure center and each point represents 1 hour of data from an ASOS station color-coded by the associated liquid equivalent snowfall rate. Points are plotted with partial transparency to avoid obscuration.





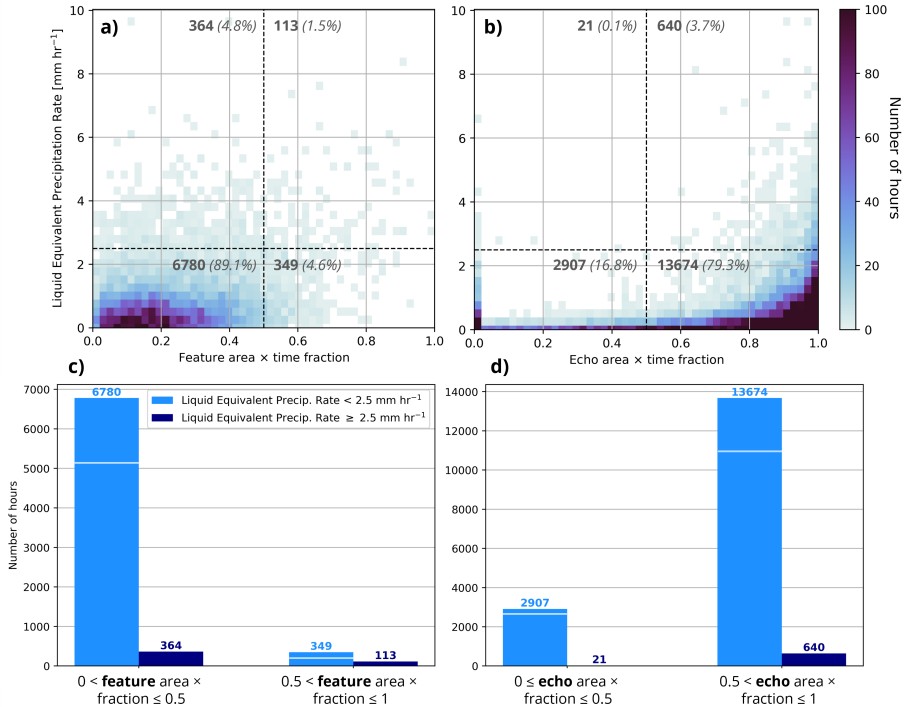

**Figure 8.** Joint distributions of number of hours for (a) all feature (faint + strong) area × time fraction and (b) all echo (background + faint features + strong features) area × time fraction versus liquid equivalent precipitation rate [mm hr$^{-1}$] for snow observations. (c) Bar chart summarizing feature area × time fraction distribution in (a) and (d) bar chart summarizing echo area × time fraction distribution in (b). Light blue bars in (c, d) represent liquid equivalent precipitation rate < 2.5 mm hr$^{-1}$ and dark blue bars represent liquid equivalent precipitation rates ≥ 2.5 mm hr$^{-1}$. Number of hours with liquid equivalent precipitation rates < 1 mm hr$^{-1}$ are annotated with horizontal lines on light blue bars in (c, d). Area × time fractions calculated with a 25 km radius and observations are paired with a 0-hour lag. Zero feature area × time fraction observations in (a, c) are not shown. In (a, b) 0.5 area × time fraction is annotated with a vertical black dashed line and 2.5 mm hr$^{-1}$ is annotated with a horizontal black dashed line. Bold annotated numbers in (a, b) indicate number of hours in each quadrant and italicized numbers indicate percent of total observations in each quadrant. The feature area × time fraction in (a, c) has 7606 total hours compared to the all echo area × time fraction in (b, d) with 17242 hours. The difference relates to the exclusion of hours with zero feature area × time fraction in (a, c).





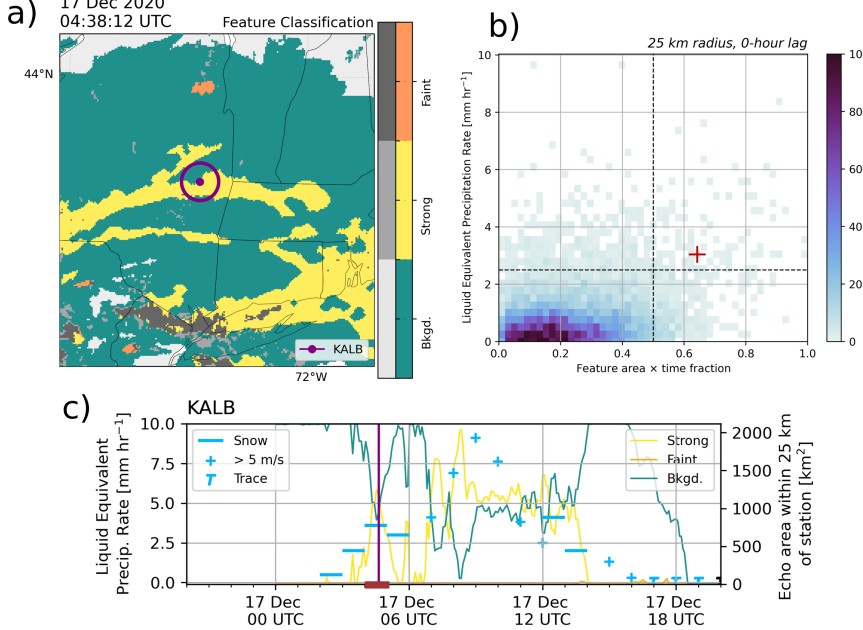

**Figure 9.** An example when a heavy snow rate ($\geq 2.5\ \mathrm{mm\ hr^{-1}}$) and high feature area $\times$ time fraction ($> 0.5$) are observed over an hour. (a) Feature detection field from NEXRAD regional mosaic at 17 December 2020 04:38:12 UTC with Albany, NY ASOS station (KALB) and 25 km radius annotated in purple, (b) 2D distribution from Fig. 8a with the specific hourly observation (04:00–05:00 UTC) annotated with red plus sign, and (c) time series of hourly precipitation rate over the entire event (16 December 20:00 UTC to 17 December 20:00 UTC) from KALB (blue annotations) and area of each feature category within 25 km of KALB (yellow: strong area, orange: faint area, and teal: background area). In (c), purple vertical line indicates time of specific NEXRAD regional mosaic in (a) and red bar on x-axis indicates the hour of observation at the red plus sign in (b). An animated version of this figure is available in the Video Supplement Animation-Figure-9.



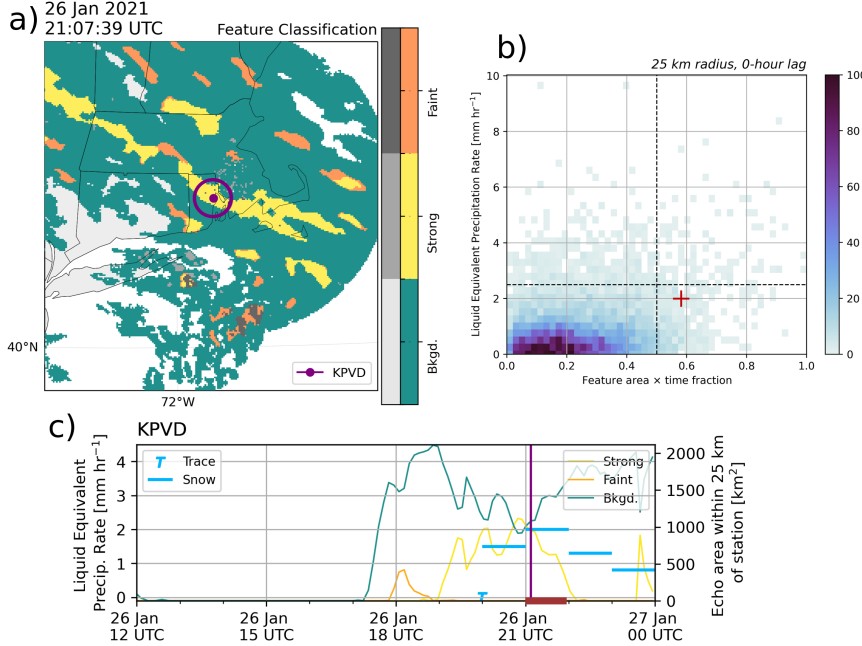

**Figure 10.** An example when a low/moderate snow rate ($< 2.5$ mm hr$^{-1}$) and high feature area $\times$ time fraction ($> 0.5$) are observed over an hour. (a) Feature detection field from NEXRAD regional mosaic at 26 January 2021 21:07:39 UTC with Providence, RI ASOS station (KPVD) and 25 km radius annotated in purple, (b) 2D distribution from Fig. 8a with the specific hourly observation (21:00-22:00 UTC) annotated with red plus sign, and (c) time series of hourly precipitation rate over the entire event (26 January 12:00 UTC to 27 January 00:00 UTC) from KPVD (blue annotations) and area of each feature category within 25 km of KPVD (yellow: strong area, orange: faint area, and teal: background area). In (c), purple vertical line indicates time of specific NEXRAD regional mosaic in (a) and red bar on x-axis indicates the hour of observation at the red plus sign in (b). An animated version of this figure is available in the Video Supplement Animation-Figure-10.





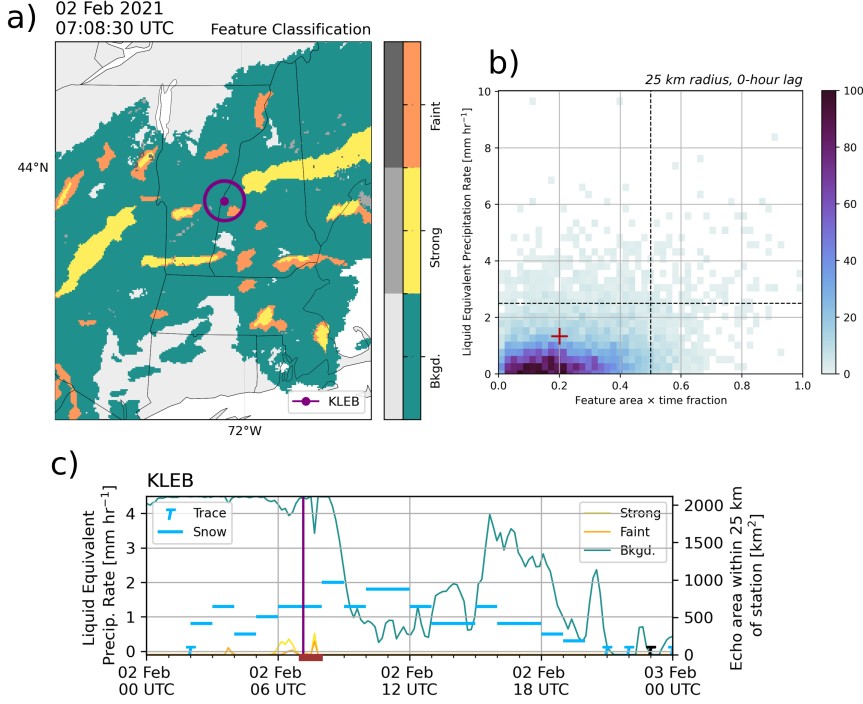

**Figure 11.** An example when a low/moderate snow rate ($< 2.5$ mm hr$^{-1}$) and low feature area $\times$ time fraction ($\leq 0.5$) are observed over an hour. (a) Feature detection field from NEXRAD regional mosaic at 02 February 2021 07:08:30 UTC with Lebanon, NH ASOS station (KLEB) and 25 km radius annotated in purple, (b) 2D distribution from Fig. 8a with the specific hourly observation (07:00-08:00 UTC) annotated with red plus sign, and (c) time series of hourly precipitation rate over the entire event (02 February 00:00 UTC to 03 February 00:00 UTC) from KLEB (blue annotations) and area of each feature category within 25 km of KLEB (yellow: strong area, orange: faint area, and teal: background area). In (c), purple vertical line indicates time of specific NEXRAD regional mosaic in (a) and red bar on x-axis indicates the hour of observation at the red plus sign in (b). An animated version of this figure is available in the Video Supplement Animation-Figure-11.



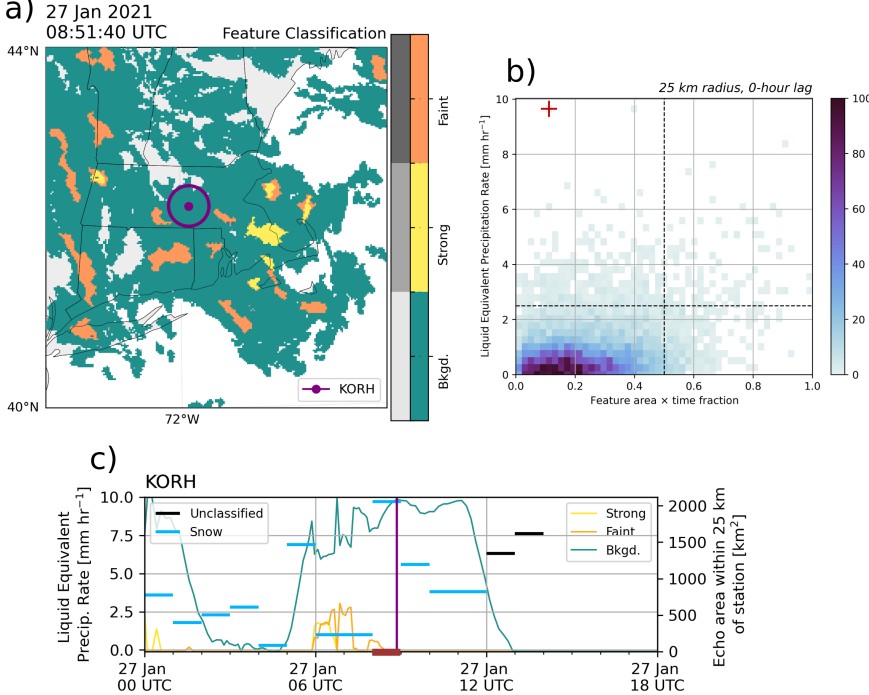

**Figure 12.** An example when a heavy snow rate ($\geq 2.5$ mm hr$^{-1}$) and low feature area $\times$ time fraction ($\leq 0.5$) are observed over an hour. (a) Feature detection field from NEXRAD regional mosaic at 27 January 2021 08:51:40 UTC with Worcester, MA ASOS station (KORH) and 25 km radius annotated in purple, (b) 2D distribution from Fig. 8a with the specific hourly observation (08:00-09:00 UTC) annotated with red plus sign, and (c) time series of hourly precipitation rate over the entire event (27 January 00:00 UTC to 27 January 18:00 UTC) from KORH (blue annotations) and area of each feature category within 25 km of KORH (yellow: strong area, orange: faint area, and teal: background area). In (c), purple vertical line indicates time of specific NEXRAD regional mosaic in (a) and red bar on x-axis indicates the hour of observation at the red plus sign in (b). An animated version of this figure is available in the Video Supplement Animation-Figure-12.



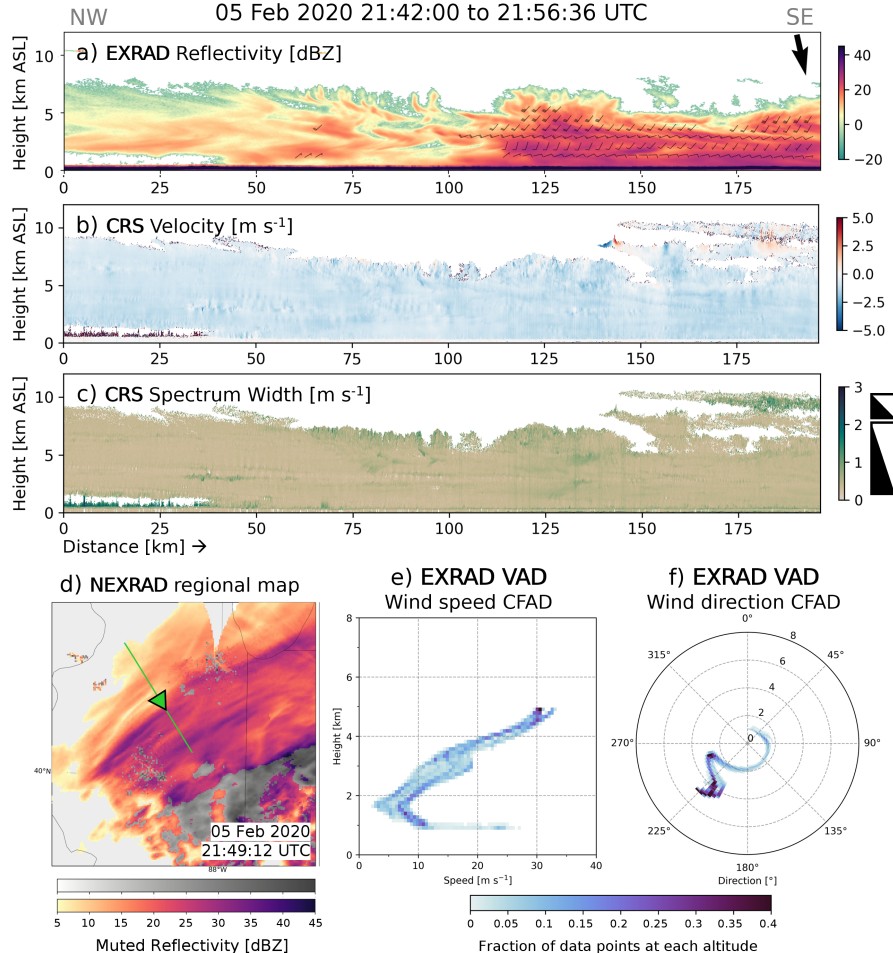

**Figure 13.** Vertical cross-section from 5 February 2020 21:42:04 to 21:56:44 UTC of (a) reflectivity [dBZ] from NASA EXRAD radar (nadir beam) and VAD winds derived from EXRAD radar (scanning beam), (b) velocity [m s$^{-1}$], and (c) spectrum width [m s$^{-1}$] from NASA CRS cloud radar. All vertical cross-sections are plotted with a 3:1 aspect ratio. Triangle icons next to (c) illustrate a 45° angle in a 1:1 and 3:1 aspect ratio. (d) Corresponding NEXRAD regional map of image muted reflectivity [dBZ] with ER-2 flight path in green, arrowhead denotes direction and location of aircraft at time of region map. CFADs of (e) wind speed and (f) wind direction of VAD winds in (a). Wind direction CFAD is plotted in polar coordinates where the angle represents the direction and each radius represents the altitude (0 km at the center). Black arrow in (a) indicates the compass direction of the aircraft during the transect.



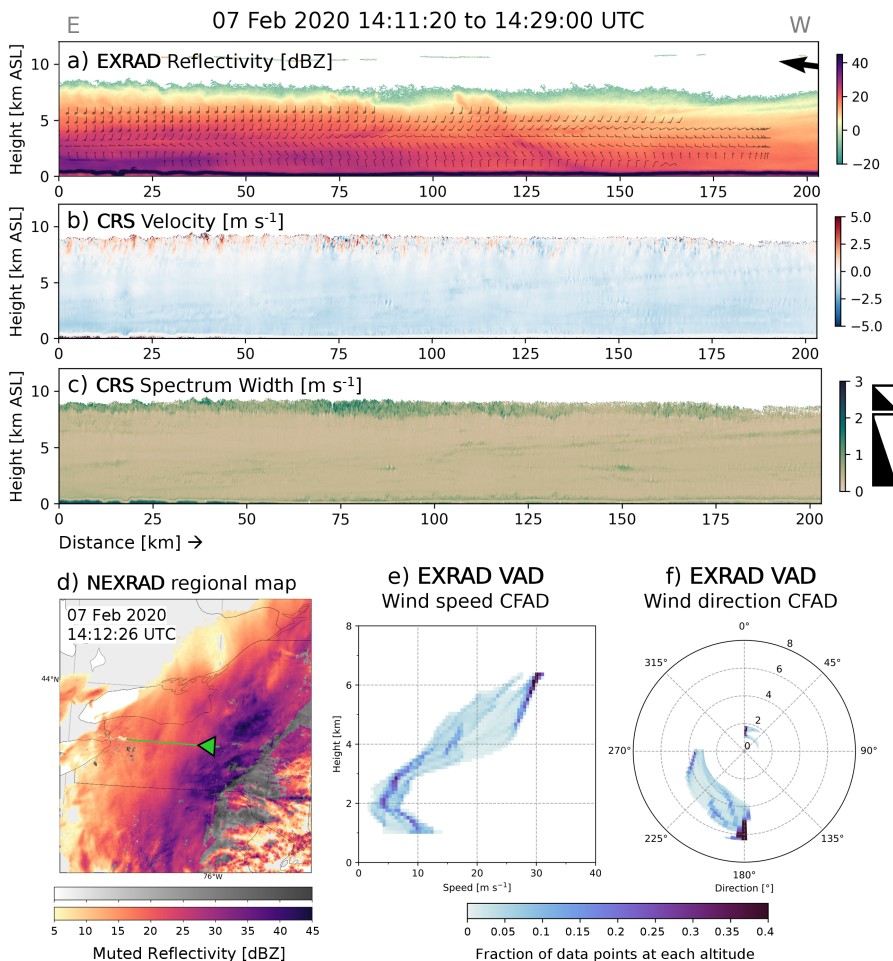

**Figure 14.** Vertical cross-section from 7 February 2020 14:11:20 to 14:29:00 UTC of (a) reflectivity [dBZ] from NASA EXRAD radar (nadir beam) and VAD winds derived from EXRAD radar (scanning beam), (b) velocity [m s$^{-1}$], and (c) spectrum width [m s$^{-1}$] from NASA CRS cloud radar. All vertical cross-sections are plotted with a 3:1 aspect ratio. Triangle icons next to (c) illustrate a 45° angle in a 1:1 and 3:1 aspect ratio. (d) Corresponding NEXRAD regional map of image muted reflectivity [dBZ] with ER-2 flight path in green, arrowhead denotes direction and location of aircraft at time of region map. CFADs of (e) wind speed and (f) wind direction of VAD winds in (a). Wind direction CFAD is plotted in polar coordinates where the angle represents the direction and each radius represents the altitude (0 km at the center). Black arrow in (a) indicates the compass direction of the aircraft during the transect.





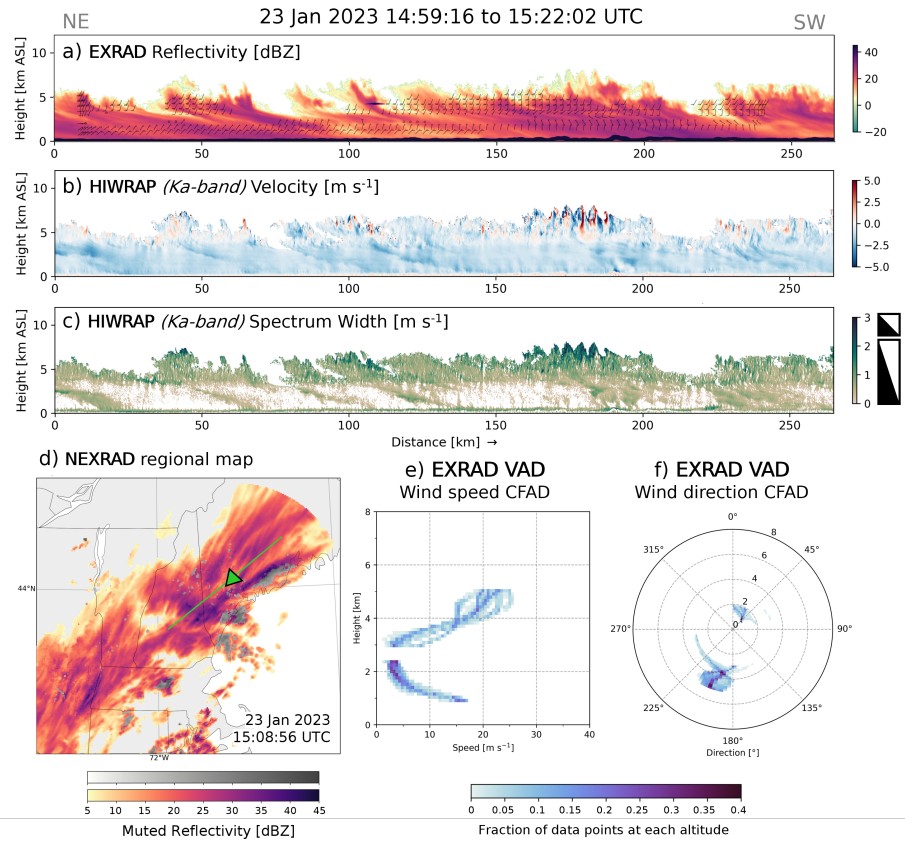

**Figure 15.** Vertical cross-section from 23 January 2023 14:59:16 to 15:22:02 UTC of (a) reflectivity [dBZ] from NASA EXRAD radar (nadir beam) and VAD winds derived from EXRAD radar (scanning beam), (b) velocity [m s$^{-1}$], and (c) spectrum width [m s$^{-1}$] from NASA HIWRAP (Ka-band) radar. All vertical cross-sections are plotted with a 3:1 aspect ratio. Triangle icons next to (c) illustrate a 45° angle in a 1:1 and 3:1 aspect ratio. (d) Corresponding NEXRAD regional map of image muted reflectivity [dBZ] with ER-2 flight path in green, arrowhead denotes direction and location of aircraft at time of region map. CFADs of (e) wind speed and (f) wind direction of VAD winds in (a). Wind direction CFAD is plotted in polar coordinates where the angle represents the direction and each radius represents the altitude (0 km at the center).



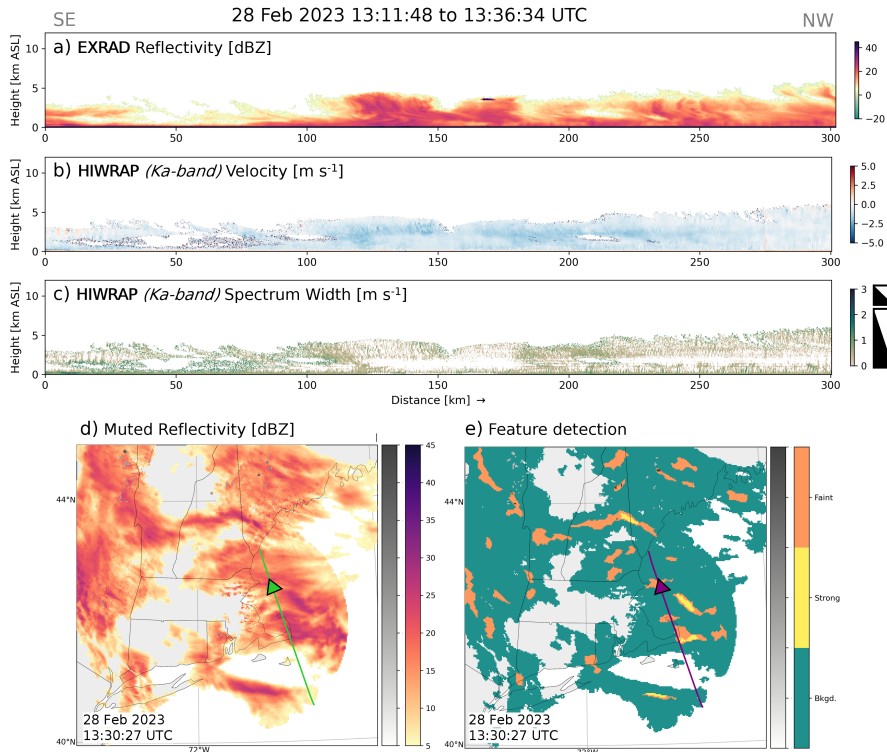

**Figure 16.** Vertical cross-section from 28 February 2023 13:11:48 to 13:36:34 UTC of (a) reflectivity [dBZ] from NASA EXRAD radar (nadir beam), (b) velocity [m s$^{-1}$], and (c) spectrum width [m s$^{-1}$] from NASA HIWRAP (Ka-band) radar. All vertical cross-sections are plotted with a 3:1 aspect ratio. Triangle icons next to (c) illustrate a 45° angle in a 1:1 and 3:1 aspect ratio. Corresponding NEXRAD regional map of (d) image muted reflectivity [dBZ] with ER-2 flight path in green and (e) feature detection classification with ER-2 flight path in purple, arrowhead denotes direction and location of aircraft at time of region map.



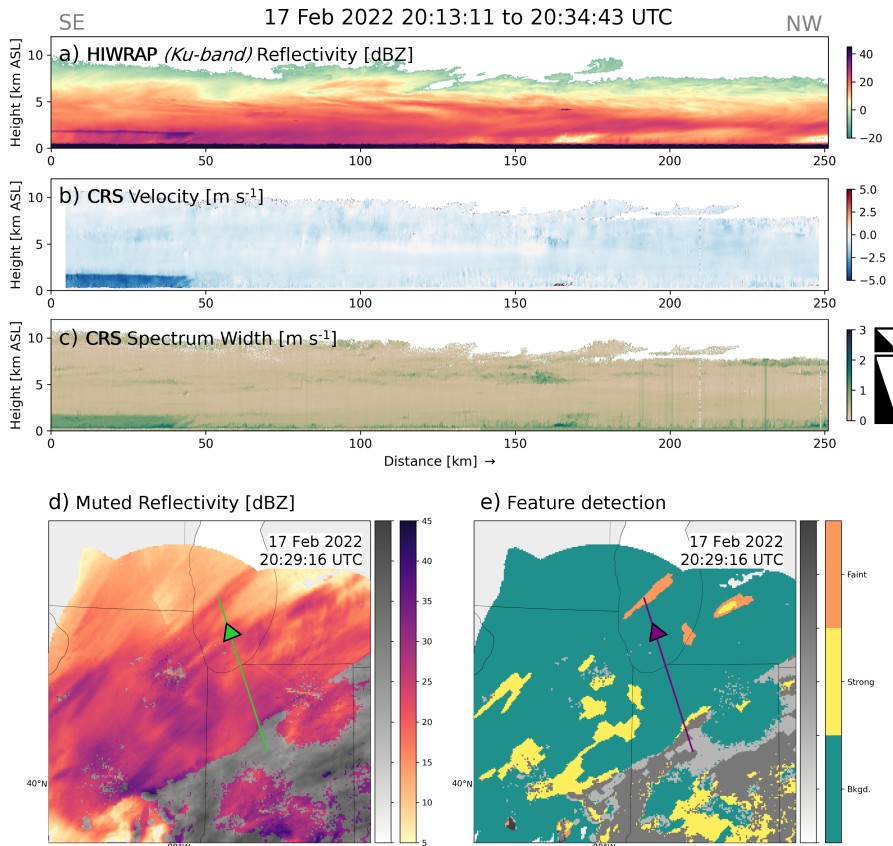

**Figure 17.** Vertical cross-section from 17 February 2022 20:13:11 to 20:34:43 UTC of (a) reflectivity [dBZ] from NASA HIWRAP (Ku-band) radar, (b) velocity [m s$^{-1}$], and (c) spectrum width [m s$^{-1}$] from NASA CRS cloud radar. All vertical cross-sections are plotted with a 3:1 aspect ratio. Triangle icons next to (c) illustrate a 45° angle in a 1:1 and 3:1 aspect ratio. Corresponding NEXRAD regional map of (d) image muted reflectivity [dBZ] with ER-2 flight path in green and (e) feature detection classification with ER-2 flight path in purple, arrowhead denotes direction and location of aircraft at time of region map.





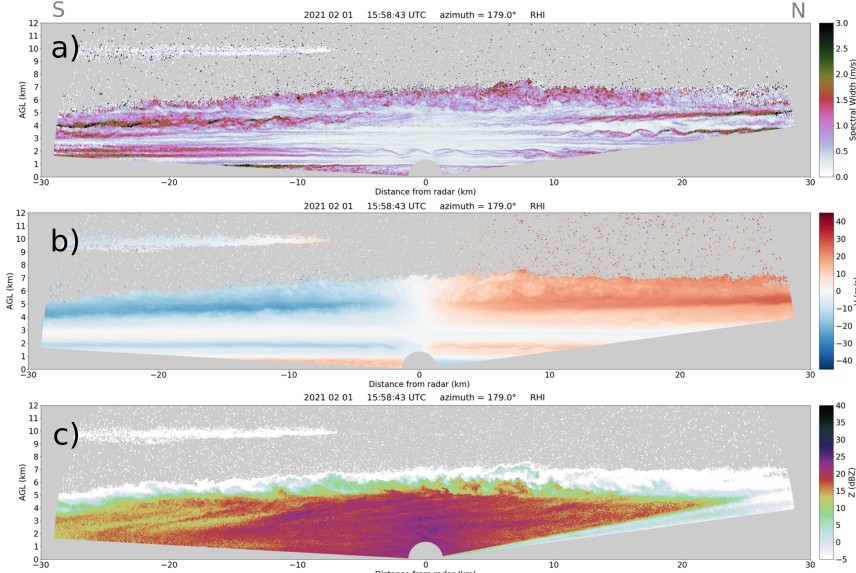

**Figure 18.** (a) Spectrum Width [m s$^{-1}$], (b) Doppler Velocity [m s$^{-1}$], and (c) Reflectivity [dBZ] RHIs from KaSPR radar at Stonybrook University at 15:58:42 UTC on 1 February 2021. RHIs are scanned up and over the radar (at 0 km on x-axis). Radar beam is partially blocked near edge of scan on right side. Plotted in a 1:1 aspect ratio. An animated version of this figure between 15:05-17:03 UTC is available in the Video Supplement Animation-Figure-18. An animation of the full storm is available in the Video Supplement Animation-Figure-18-19-full.



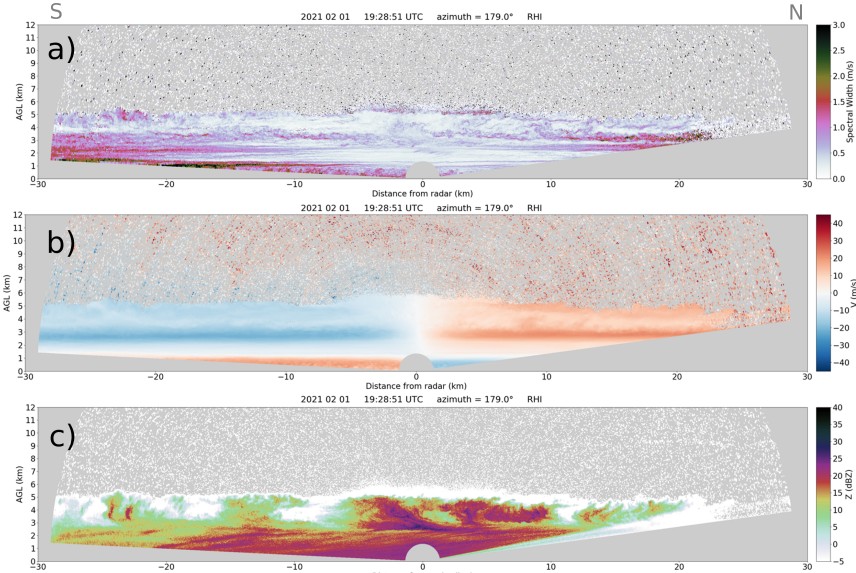

**Figure 19.** (a) Spectrum Width $[\mathrm{m\,s^{-1}}]$, (b) Doppler Velocity $[\mathrm{m\,s^{-1}}]$, and (c) Reflectivity $[\mathrm{dBZ}]$ RHIs from the KaSPR radar at Stonybrook University at 19:28:51 UTC on 1 February 2021. The RHIs are scanned up and over the radar (at 0 km on x-axis). Radar beam is partially blocked near edge of scan on right side. Plotted in a 1:1 aspect ratio. An animated version of this figure between 18:24-20:25 UTC is available in the Video Supplement Animation-Figure-19. An animation of the full storm is available in the Video Supplement Animation-Figure-18-19-full.



**Table 1.** Table of microphysical processes and their associated change to mass per unit volume (IWC/LWC) and to radar reflectivity.

| Process | Change to IWC/LWC | Change to radar reflectivity |
|---|---|---|
| Riming | Increase | Increase |
| Vapor Deposition | Increase | Increase |
| Collision-Coalescence | Increase | Increase |
| Condensation | Increase | Increase |
| Aggregation | *No change* | Increase |
| Melting | *No change* | Increase |
| Evaporation | Decrease | Decrease |
| Sublimation | Decrease | Decrease |
| Freezing | *No change* | Decrease |
| Fragmentation | *No change* | Decrease |
| Raindrop Breakup | *No change* | Decrease |



**Table 2.** Band, frequency [GHz], sensitivity [dBZ] at 10 km, horizontal spatial resolution [m] at 10 km below the aircraft and vertical spatial resolution [m], and citation for radars deployed on the ER-2 aircraft during NASA IMPACTS. Band, frequency [GHz], sensitivity [dBZ] at 1 km, horizontal spatial resolution [m] at 10 km above the radar and vertical spatial resolution [m], and citation for KaSPR deployed at Stonybrook University during NASA IMPACTS.

| Radars | Band | Frequency | Sensitivity | Spatial Resolution (H, V) | Beamwidth | Reference |
|---|---|---|---|---|---|---|
| **EXRAD** *(precip. radar)* | X | 9.6 GHz | -9 dBZ | 679 m, 110 m | 3.4° | Heymsfield et al. (1996) |
| **HIWRAP** *(precip. radar)* | Ku | 13.5 GHz | -17 dBZ | 672 m, 150 m | 3.0° | Li et al. (2016) |
| | Ka | 35.5 GHz | -17 dBZ | 288 m, 150 m | 1.2° | |
| **CRS** *(cloud radar)* | W | 96 GHz | -34 dBZ | 137 m, 115 m | 0.64° | McLinden et al. (2021) |
| **KASPR** *(precip. radar)* | Ka | 35.3 GHz | -40 dBZ | 56 m, 15 m | 0.32° | Oue et al. (2024) |