# Peer review of "Synthesis of surface snowfall rates and radar-observed storm structures in 10+ years of Northeast US winter storms"

_EGUsphere, 2025_

## Author Comment (AC1)

**Response to Reviewer #1**

***Author reponses are black, bolded, and italicized***
***Text updated in the paper are green, bolded, and italicized***

The topic of relating observed surface snowfall rates to local enhancements in radar reflectivity is relevant, interesting, and well within the scope of ACP. However, in ACP's Aim & Scope it is mentioned that "articles with a local focus must clearly explain how the results extend and compare with current knowledge." It is not clear how the local focus (NE U.S.) could be extended/applied to non-orographic snowfall in other parts of the word, so the paper would benefit from some additional context.

***Snow bands have been documented in many regions outside of the US including Europe (Mazon et al. (2015), Norris et al. (2013)), Asia (Fujiyoshi et al. (1998), Murakami (2019), Sato et al. (2022), Zhao et al. (2020)), and the Southern Hemisphere (Comin et al. (2018)). The snowbanding literature from Europe and Japan has primarily focused on lake/sea-effect snow banding (Mazon et al. (2015), Norris et al. (2013), Fujiyoshi et al. (1998), Murakami (2019), Sato et al. (2022)). Snowbanding in the Southern Hemisphere in Chile and Argentina is orographic in nature and is not relevant to this work (Comin et al. (2018)). Winter storms in eastern Asia (China, Russia, Korea) can exhibit banded features, and this work could be extended to these types of storms (Zhao et al. (2020)).***

***We have added the following to the Introduction (line 50):***

***"While most of the snow band literature has focused on the northeast US, non-orographic snow bands have been documented in winter storms across the world. Lake/sea-effect snow banding has been investigated in Europe and Japan (Mazon et al., 2015; Norris et al., 2013; Fujiyoshi et al., 1998; Murakami, 2019; Sato et al., 2022). Winter storms in eastern Asia (China, Russia, Korea) also exhibit banded features (Zhao et al., 2020)."***

- ***Comin, Alcimoni Nelci, et al. "Impact of different microphysical parameterizations on extreme snowfall events in the Southern Andes." Weather and climate extremes 21 (2018): 65-75.***

- *Fujiyoshi, Yasushi, Naohiro Yoshimoto, and Takao Takeda. "A dual-Doppler radar study of longitudinal-mode snowbands. Part I: A three-dimensional kinematic structure of meso-γ-scale convective cloud systems within a longitudinal-mode snowband." Monthly weather review 126.1 (1998): 72-91.*
- *Mazon, Jordi, et al. "Snow bands over the Gulf of Finland in wintertime." Tellus A: Dynamic Meteorology and Oceanography 67.1 (2015): 25102.*
- *Murakami, Masataka. "Inner structures of snow clouds over the Sea of Japan observed by instrumented aircraft: A review." Journal of the Meteorological Society of Japan. Ser. II 97.1 (2019): 5-38.*
- *Norris, Jesse, Geraint Vaughan, and David M. Schultz. "Snowbands over the English Channel and Irish Sea during cold‐air outbreaks." Quarterly Journal of the Royal Meteorological Society 139.676 (2013): 1747-1761.*
- *Sato, Kazutoshi, Takao Kameda, and Tatsuo Shirakawa. "Heavy snowfall at Iwamizawa influenced by the Tsushima warm current." Journal of the Meteorological Society of Japan. Ser. II 100.6 (2022): 873-891.*
- *Zhao, Yu, et al. "Case study of a heavy snowstorm associated with an extratropical cyclone featuring a back-bent warm front structure." Atmosphere 11.12 (2020): 1272.*

This work presents relatively extensive and comprehensive analyses of snowfall data spanning 11 winter seasons and 264 storms days, leading to some novel findings about the relationship between the spatial distribution of enhanced radar echoes and associated snowfall amounts at the surface. The primary conclusion is that—contrary to more limited case studies focusing on extreme snowfall events in the northeastern U.S.—high rates of snowfall do not necessarily correspond to locally enhanced reflectivity features (i.e., mesoscale snow bands) in a large dataset such as this. In fact, only about one out of every four hourly occurrences have heavy surface snow rates when these enhanced reflectivity features pass over a site. This conclusion is indeed substantial.

The methods are laid out in a way that is mostly clear and easy to follow, although there are a few areas highlighted in my comments below added clarity is needed. All assumptions appear reasonable and results are thoroughly laid out and explained in a good amount of detail, leading to the conclusions in a logical manner. The paper is well-organized and well-written, overall.

**Specific comments:**

1. Abstract: It would be helpful to include one or two sentences about the purpose of this study. Perhaps something like 'relationship between multi-bands and frontogenesis is not clear' from lines 30-31.

   **We have added the following text to the abstract (line2):**

   **"While primary bands are associated with frontogenesis, multi-bands are found in environments with both frontogenesis and frontolysis."**

2. Line 63: I'm not sure what you mean by 'better than a factor of 2'? Is this based on some prior work that you could cite here?

   **We have rephrased the text and added a reference to the figure as follows (line 76):**

   **"For reflectivities > 0 dBZ, which usually contain some precipitation-size falling ice particles, IWC generally increases as Z increases. But given the spread of the observed values, there is at least a factor of two uncertainty in the volumetric ice mass as a function of radar reflectivity (Fig. 1; Zaremba et al., 2023)."**

   **The figure below shows coincident observations from 1 Hz Nevzorov probe Ice Water Content samples and Wyoming Cloud Radar reflectivities obtained in light snow by the University of Wyoming King Air during the SNOWIE project in the mountains of western Idaho. I annotated some red bars to indicate that for given reflectivity values, the spread of IWC values is such that we cannot quantify the IWC more accurately than within a range that spans a factor of 2.**

[Figure]

3. Line 67, 78: Is there no lake-effect snow in upstate New York? Any reason to believe that KALB and KGBM wouldn't be impacted by lake-effect snow (especially KGBM)? What is the reasoning for excluding lake-effect snow and how important is it that the dataset isn't contaminated by these events?

*Yes, lake-effect events have banded features and impact this general region but lake-effect snow bands are very different from mesoscale snow bands produced during mid-latitude snow storms which is outside the focus of this study. It is possible that KBGM and KALB would allow for the inclusion of a few lake event storms, however, the majority of our events include a station closer to the coast. If our sample of events included a lot of lake-event cases, it could potentially bias our results in the direction of a strong relationship between banding and surface snowfall since lake-effect snow bands may have a stronger relationship to surface snow rates.*

*We added the following to the paper (line 92):*

*"While the majority of the stations we use to define winter storm events are close to the coast, it is possible that by using the Albany, NY and Binghamton, NY stations that some lake-effect events are included in our dataset. Lake-effect snow bands*

*passing overhead have a documented relationship to heavier snow rates (Kristovich et al., 2017; Kosiba et al., 2019; Mulholland et al., 2017) so their inclusion in the data set could slightly bias the results toward a stronger relationship between snowbands and heavier surface snow than is true for stations further from the Great Lake shores."*

4. Line 83: What is difference b/w 29 ASOS stations and 14 stations of GHCNd? Are the 14 GHCNd stations simply for determining dates/times of non-orographic winter storm events? What measurements were included? Just daily snow depth?

*The GHCNd database produces daily summaries from some surface stations (some of which are ASOS stations). The daily summaries include snow depth which are used to define our events. We did not use any other information from this dataset. I rephrased the text to clarify (line 89):*

*"The daily snow depth data at each station was gathered from the Global Historical Climatology Network daily (GHCNd) database (Menne et al., 2012)."*

5. Line 91: Do you have a reference for 'do not have a heated rim and thus are subject to capping'? It seems this could be a significant problem that could use more discussion here. For example, are there times that this could be happening and resulting in an underestimation of the snow accumulation? Is it possible that it could happen and report lower-than-actual amounts of snow (rather than no snow at all)?

*We included a citation to the Martinaitis text which discusses heated rims in AWPAG sensors. We have rephrased the text to the following (line 112):*

*"AWPAG sensors do not have a heated rim and thus are subject to capping, although it is difficult to estimate how often this occurs in our dataset (Martinaitis et al., 2015). If capping does occur, no snow would be reported for an hour, resulting in an underestimation of the snow accumulation for that hour. Partial capping can also occur and would result in an underestimation of snow for a period of time followed by an overestimate once the cap releases (Baker Perry, personal communication)."*

6. Line 95: Is a collection efficiency of less than one something that you correct/account for? If so, how? If not, why is it not necessary?

*Collection efficiency is not something that we can directly consider with this dataset which is why we use the wind speed threshold instead.*

7. Line 114: Is the interpolation technique described in detail elsewhere? I see later on that you cite Tomkins et al. (2022) for details. Can you add a few words to describe the technique? Is it linear interpolation? Or is it too lengthy/complicated to mention here?

*We are using Cressman weighting in the interpolation. We have included the reference and pointed readers to the Tomkins et al. (2022) citation. Text rephrased as follows (line 137):*

*"We interpolate the single elevation to yield a 2D grid using Cressman weighting (Cressman, 1959, for full details see Tomkins et al. (2022))."*

8. Lines 321-322, 325, 327: It has been a little while since I've looked at a VAD wind profile. When you say 'lots of wind shear illustrated by the wind barbs'—please describe what the direction is showing (e.g., any vertical component or horizontal only?). Are there higher wind speeds in the CFAD in 13e that aren't seen in 13a? I think a little bit more description of what we are looking at for wind direction would help here.

*The VADs show a distance-height profile of horizontal winds only, no vertical component. The data used to create the CFADs in Fig. 13 e,f are the same as what is plotted in Fig. 13a. We have added the following text to clarify (line 368):*

*"The wind barbs in the VAD profiles indicate the direction and speed that the wind is coming from and correspond well to the patterns in the reflectivity field. The CFADs summarize the distributions with height of direction and speed of the winds in the VAD profiles along the flight leg."*

9. Line 324: For the triangle icons it would be helpful to see a version of one of these images with a 1:1 aspect ratio—perhaps rotated to fit and place in the supplemental? Or just a subset perhaps. This would be nice to reference and easier to see how this would change streamer angles as you describe.

*We have added the following image to the supplemental material to illustrate the difference between using a 3:1 (top) and 1:1 (bottom) aspect ratio and how the 3:1 aspect ratio distorts some of the features:*

[Figure]

10. Line 336: There is no aircraft vector in 15a like there is in 13a & 14a.

 *We have updated the figures. Thank you for noticing this.*

11. Line 338: Can you comment on features/scales resolved by some of these radars & not by others of note here, and how you would or would not expect that to influence the interpretation?

*We have added the following text (line 237):*

*"EXRAD and HIWRAP are precipitation radars while the CRS is a cloud radar. The cloud radar is more sensitive to smaller ice particles than the precipitation radars and has a smaller radar resolution volume size which is capable of observing finer-scale features (see Table 2 for details including the resolution of each radar at 10 km distance)"*

12. Line 342: It didn't become obvious to me until this line that the 3:1 aspect ratio is not image aspect ratio but distance aspect ratio (I think because I am used to seeing

aspect ratio used with respect to image dimensions rather than physical dimensions). Perhaps it is not just me and this could be made clear the first time it is mentioned.

**We have rephrased the text to clarify (line 370):**

*"It is important to note that the cross sections are plotted in a 3:1 aspect ratio (3 units in vertical to 1 unit in horizontal distance) so features that are tilted appear to be more vertical in the plots (see triangle icons next to Fig. 13c for visualization). An example transect showing a comparison between a 1:1 aspect ratio and a 3:1 aspect ratio is available in the Supplementary Material."*

13. Line 344: The streamers appear to be streaming in direction opposite that of wind flow (i.e., the same direction that the aircraft is heading in). Or perhaps I am misunderstanding the wind flow direction or the orientation of the streamers?

**In this case the wind barbs indicate the direction that the wind is coming from (opposite to the direction the aircraft is heading). We have rephrased the text to the following (line 368):**

*"The wind barbs in the VAD profiles indicate the direction and speed that the wind is coming from and correspond well to the patterns in the reflectivity field. The CFADs summarize the distributions with height of direction and speed of the winds in the VAD profiles along the flight leg. It is important to note that the cross sections are plotted in a 3:1 aspect ratio (3 units in vertical to 1 unit in horizontal distance) so features that are tilted appear to be more vertical in the plots (see triangle icons next to Fig. 13c for visualization)."*

14. Line 429: Is there any physical/dynamical reasoning to support the observation that the highest snow rates tend to be in the northwest and northeast quadrants?

**We have added the following paragraph (line 224):**

*"The northwest quadrant typically contains the occluded front and northeast quadrant typically contains the upward sloping frontal boundary ahead of the warm front. Frontogenesis is typically present in both the NE and NW quadrants (Novak et al., 2004). Additionally, for northeast US winter storms the near surface air in these*

*quadrants is more often cold enough to support snow as compared to the southwest quadrant cold frontal region that drapes further south and often yields rain."*

**Relevant figure from Novak et al. (2004):**

[Figure]

15. Figure 5: I think it is worth it to explicitly mention here that blue horizontal lines correspond to left-hand side (or coloring the left-hand vertical axis blue?)

**We have rephrased part of the Fig. 5 caption as follows:**

*"(c) The corresponding time series of ASOS hourly precipitation rate (left axis) valid from 7 February 2021 06 UTC to 08 February 2021 01 UTC and echo areas (right axis) calculated within 25 km of Boston, MA ASOS station (KBOS: red dot in (a) and purple dot in (b))."*

16. Figure 6: I'm not sure if a) is adding much value. Is it more useful to show the intensity of the low centers than the tracks? Perhaps color-code beginning to end of tracks rather than pressure. Should we be seeing any kind of pattern in the intensity of these low centers? If not then consider omitting or coloring based on beginning to end (e.g., lighter at beginning to dark at end of path).

**Yes, we understand that this plot is not the easiest to see but we wanted to include it to provide context for the density plot in (b). We are not expecting the reader to interpret any patterns in the pressure values for this plot.**

17. Figure 9 (and possibly others): You don't mention what the gray colors are here. These are where you removed melting/mixed pixels?

*We have added the following to the caption of Figs. 9-12:*

*"In (a), gray colors indicate regions of partially-melted or mixed precipitation."*

18. Figure 13: CFADs of (e) wind speed and (f) wind direction; this wind direction in (f) is the direction *from* which the wind is blowing, correct?

*We rephrased part of the caption of Figs. 13-15 to read:*

*"Wind direction CFAD indicates the direction the wind is coming from and is plotted in polar coordinates where the angle represents the direction and each radius represents the altitude (0 km at the center)."*

19. Figure 14: Perhaps just say "Same as Figure 13 except for…"

*We intentionally repeat the captions for each figure to avoid confusion for the reader.*

20. Table 1: How is change to mass per unit volume (IWC/LWC) determined? Is this IWC/LWC a ratio or are you indicating an increase to one or the other or both?

*We are listing the change to mass per unit volume as either IWC in the case of frozen precipitation and LWC in the case of liquid precipitation. We have clarified the Table 1 caption as follows to be more clear about this:*

*"Table of microphysical processes and their associated change to mass per unit volume (IWC and LWC), and radar reflectivity. Radar reflectivity is a function of diameter and dielectric constant.The dielectric constant is larger for liquid water than for ice particles. "*

*We have also included an expanded version of the table here with 2 extra columns, change to particle diameter and change to dielectric constant to provide more context for the change to mass per unit volume. In the paper, we have included the "Change to particle diameter" column.*

| Process | Change to mass per unit | Change to radar | Change to particle | Change to dielectric |
|---|---|---|---|---|

|  | volume (IWC and LWC) | reflectivity | diameter | constant |
|---|---|---|---|---|
| Riming | Increase | Can increase [1,3] | Can increase [1,3] | Slight increase[2] |
| Vapor Deposition | Increase | Usually increase[1] | Increase | Slight change[2] |
| Collision-Coalescence | Increase | Increase | Increase | *No change* |
| Condensation | Increase | Increase | Increase | *No change* |
| Aggregation | *No change* | Increase | Increase | Slight decrease[2] |
| Melting | *No change* | Increase | Usually decrease[1] | Increase |
| Evaporation | Decrease | Decrease | Decrease | *No change* |
| Sublimation | Decrease | Usually decrease[1] | Decrease | Slight change[2] |
| Freezing | *No change* | Decrease | Usually increase[1] | Decrease |
| Ice Fragmentation | *No change* | Decrease | Decrease | Slight increase[2] |
| Raindrop breakup | *No change* | Decrease | Decrease | *No change* |

**[1] Depends on ice crystal habit or habits of preexisting precipitation particle**
**[2] Changes in dielectric constant related to changes ice density are much smaller than those associated with change of phase**
**[3] Depends on degree of riming. For example, light riming will not change reflectivity or diameter much if at all.**

Technical/typing corrections:

21. Abstract (line 7): 'echo' should be plural 'echoes'

*Done.*

22. Line 60: I believe it should be hyphenated 'surface-snow-producing'

*Done.*

23. Line 110: NWS acronym should be defined upon first use in Line 43

*Done.*

24. Line 199: 'in terms joint' should be 'in terms of joint'?

   ***Done.***

25. Line 299: ground-based-scanning-radar-observed (I think would be the proper hyphenation here)

   ***Done.***

26. Line 306: snow form should be 'snow to form'?

   ***Done.***

27. Line 337: I think this is supposed to 'Fig. 15' not 'Fig. 15e'

   ***Done.***

28. Line 379: 'is' should be 'are'

   ***Done.***

29. Line 380: 'how a individual features' should be 'how a storm's individual features'?
   ***Done.***

---

## Author Comment (AC2)

**Response to Reviewer #2**

***Author reponses are black, bolded, and italicized***
***Text updated in the paper are green, bolded, and italicized***

This study investigates an important topic regarding how to improve the quantification of surface snowfall rate from radar reflectivity. The authors applied a large amount of objectively analyzed data from NWS operation radars and airborne and ground-based radars from a field campaign, as well as ASOS data. I find that the analysis is unique and logic, and the conclusions are mostly reasonable. However, I would like to see more clarifications, such as the interpretation between the defined feature and the continuous radar reflectivity values. Some other comments are given below as well.

**Specific comments:**

1. Line 31: The presence of frontolysis in snowbands was analyzed in Han et al. (2007) through solving the Sawyer Eliassen equation. It is relevant to the "frontolysis" discussion here. Please refer to the study and its frontolysis anlysis.

   ***We added the following sentence (line 37) :***

   ***"Han et al. (2007) examined the precipitation structure of 2 winter storms and found that couplets of frontogenesis and frontolysis were present in the vicinity of both the occluded front and the warm front"***

2. Table 1: It is a good idea to consider the microphysical processes and the difference between the mass and radar reflectivity. But please put in justification. A thorough justification may have to consider radiative transfer calculation, which is not necessary if the authors do not already have those experience or knowledge. But some degree of justification is necessary.

   ***(Repeated from Reviewer 1 response)***

   ***We are listing the change to mass per unit volume as either IWC in the case of frozen precipitation and LWC in the case of liquid precipitation. We have clarified the Table 1 caption as follows to be more clear about this:***

*"Table of microphysical processes and their associated change to mass per unit volume (IWC and LWC), and radar reflectivity. Radar reflectivity is a function of diameter and dielectric constant.The dielectric constant is larger for liquid water than for ice particles. "*

**We have also included an expanded version of the table here with 2 extra columns, change to particle diameter and change to dielectric constant to provide more context for the change to mass per unit volume. In the paper, we have included the "Change to particle diameter" column.**

| Process | Change to mass per unit volume (IWC and LWC) | Change to radar reflectivity | Change to particle diameter | Change to dielectric constant |
|---|---|---|---|---|
| Riming | Increase | Can increase [1,3] | Can increase [1,3] | Slight increase[2] |
| Vapor Deposition | Increase | Usually increase[1] | Increase | Slight change[2] |
| Collision-Coalescence | Increase | Increase | Increase | *No change* |
| Condensation | Increase | Increase | Increase | *No change* |
| Aggregation | *No change* | Increase | Increase | Slight decrease[2] |
| Melting | *No change* | Increase | Usually decrease[1] | Increase |
| Evaporation | Decrease | Decrease | Decrease | *No change* |
| Sublimation | Decrease | Usually decrease[1] | Decrease | Slight change[2] |
| Freezing | *No change* | Decrease | Usually increase[1] | Decrease |
| Ice Fragmentation | *No change* | Decrease | Decrease | Slight increase[2] |
| Raindrop breakup | *No change* | Decrease | Decrease | *No change* |

**[1] Depends on ice crystal habit or habits of preexisting precipitation particle**
**[2] Changes in dielectric constant related to changes ice density are much smaller than those associated with change of phase**
**[3] Depends on degree of riming. For example, light riming will not change reflectivity or diameter much if at all.**

3. Line 55: "Unlike convective cells …" this sentence is not clear.  I think you are probably saying that in convective cells, the snow/graupel particles aloft melt while

they fall to warmer temperature at low levels, which is the melting level. So, the enhanced reflectivity column from the frozen particles does not usually extends to the surface …… However, in snowstorms, it is different. Please make this sentence clear.

**We have rephrased the paragraph to read (line 60):**

*"There are several complicating factors in the analysis of winter storms using weather radar observations. Increases in radar reflectivity in snow do not necessarily equate to increases in mass per unit volume (Table 1). Aggregation and partial melting increase the radar reflectivity but do not change the mass per unit volume. Additionally, there are important differences between the 3D structures of enhanced reflectivities in rain versus snow. In warm-season precipitation systems the 0∘C level is 3 km altitude or more above the surface and it is reasonable to deduce that stronger locally-enhanced radar reflectivity features a few km above the surface are associated with higher rain rates at the surface. The typical fall speeds of raindrops (~2-8 m s$^{-1}$, depending on raindrop size) often yield nearly vertical columns of enhanced reflectivity features in rain layers. In contrast, the slower fall speed of snow, (~ 1 m s$^{-1}$, equivalent to 33 minutes to fall 2 km) yields sufficient time for the the advection of the snow by horizontal winds, which are typically ≥10 m s$^{-1}$, to form curved ice streamers (Wexler, 1955; Wexler and Atlas, 1959). Falling snow particles can be blown sideways more than 50 km horizontally from the locations where they first achieve precipitation size near the top of the storm."*

4. Line 63: last sentence of this paragraph. Can you please quantify the range of the factor as well?

**It is not clear what you are asking. For Reviewer #1 we have rephrased the text and added a reference to the figure as follows (line 76):**

*"For reflectivities > 0 dBZ, which usually contain some precipitation-size falling ice particles, IWC generally increases as Z increases. But given the spread of the observed values, there is at least a factor of two uncertainty in the volumetric ice mass as a function of radar reflectivity (Fig. 1; Zaremba et al., 2023)."*

5. Line 67: the key finding is not clear. I think it needs to be clarified that snow bands are instantaneous features in the radar reflectivity at a certain level above the ground. Even if it can be related to the surface snow rate, like you would argue for

rain, it is still just instantaneous snowfall rate. Or you may need to say frontogenesis-related primary band are more likely to be associated with hourly accumulated surface snowfall, but multi-bands may not.

**We have rephrased the last sentence of the introduction to read (line 80):**

*"The key finding is that locally enhanced linear features (i.e. mesoscale snow bands) in operational scanning radar reflectivity within northeast US snow storms (which exclude orographic and lake effect snow storms) are usually not associated with heavy surface snow rates."*

6. Line 130: Please provide information how the reflectivity is rescaled to snow rate. Also, please clarify that this snow rate is not liquid-equivalent as ASOS's is.

**We added the following text to clarify (line 155):**

*"To rescale the reflectivity, we use the wet snow Z-S relationship from Rasmussen et al. (2003); $Ze = 57.3S^{1.67}$ where Ze is equivalent radar reflectivity with units of $mm^6\ m^{-3}$ and S is snow rate with units of $mm\ hr^{-1}$. Note that the snow rate obtained from this relationship represents an instantaneous rate and is not liquid-equivalent unlike the data reported from the ASOS stations."*

7. Line 146 and 147: You define two metrics called 'area x time fraction'. Both are based on the definition of feature, the first one includes strong and faint features, and the second one adds in the background. The features and background are sort of masks from the radar reflectivity, which does not reflect the continuous value from the radar reflectivity. So, why not use radar reflectivity to define a metric to account for more continuous change of the radar reflectivity?

**We have added the following to Section 2.1.3 (line 163):**

*"The relationship between reflectivity and liquid equivalent snow rate has large uncertainty as it depends on the ice particle shape, density, degree of riming, aggregation, and terminal velocities of the snow particles present. It would not be suitable to use the quantitative values from a single Z-S relationship (e.g. Fujiyoshi et al., 1990; Mitchell et al., 1990; Rasmussen et al., 2003; Matrosov et al., 2008; von Lerber et al., 2017; Wen et al., 2017). We developed our technique based on the method of Krajewski et al. (1992) who used an area-threshold method to estimate*

*mean areal rainfall using radar reflectivity observations. Krajewski et al. found that when there is high uncertainty in the Z-R relationship, the rainfall estimates from the area-threshold method perform significantly better than the estimates from the Z-R relationship. Our area × time fraction method estimates space-time integrals from quantities derived from radar reflectivity spatial patterns thereby avoiding issues that would result from using a highly uncertain transformation from reflectivity to snow rate. Yeh and Colle (2025) compared several different identification methods for identifying snow bands in winter storms and found that object-based methods performed better than methods using an absolute or variable reflectivity thresholds."*

*References:*

*von Lerber, Annakaisa, et al. "Microphysical properties of snow and their link to Z e–S relations during BAECC 2014." Journal of Applied Meteorology and Climatology 56.6 (2017): 1561-1582.*

*Mitchell, David L., Renyi Zhang, and Richard L. Pitter. "Mass-dimensional relationships for ice particles and the influence of riming on snowfall rates." Journal of Applied Meteorology and Climatology 29.2 (1990): 153-163.*

*Rasmussen, Roy, et al. "Snow nowcasting using a real-time correlation of radar reflectivity with snow gauge accumulation." Journal of Applied Meteorology 42.1 (2003): 20-36.*

*Matrosov, Sergey Y., Matthew D. Shupe, and Irina V. Djalalova. "Snowfall retrievals using millimeter-wavelength cloud radars." Journal of Applied Meteorology and Climatology 47.3 (2008): 769-777.*

*Wen, Yixin, et al. "Evaluation of MRMS snowfall products over the western United States." Journal of Hydrometeorology 18.6 (2017): 1707-1713.*

*Krajewski, Witold F., et al. "The accuracy of the area-threshold method: A model-based simulation study." Journal of Applied Meteorology and Climatology 31.12 (1992): 1396-1406.*

*Yeh, Phillip, and Brian A. Colle. "A Comparison of Approaches to Objectively Identify Precipitation Structures Within the Comma Head of Mid-Latitude Cyclones." Journal of Atmospheric and Oceanic Technology (2025).*

8. Line 150: I think you mean "Fig. 5d".

*Done.*

9. Line 182: Do you mean "Fig. 6a"?

*We meant "Fig. 7". Thanks for catching this.*

10. Line 250: "equating snow bands with heavy snow will … over prediction …". I understand the goal and the method of this study. However, in Numerical Weather Prediction (NWP) practice, I don't think the snowfall prediction is based on radar reflectivity bands, neither the observed nor the simulated radar reflectivity. I think this statement needs to be modified. Please also give reference of how National Weather Service use NWP models for snowfall prediction.

*We have added the following paragraph to the introduction (line 25):*

*"US National Weather Service (NWS) forecasters use a variety of methods to predict snowfall accumulations. From numerical models, they use ice water context (IWC) as well as the presence of banded features from simulated reflectivity to estimate quantitative precipitation Radford et al. (2023). For nowcasting, NWS forecasts use observed radar reflectivity to determine locations of heavy snow (David Novak, personal communication)."*

*Reference:*

*Radford, J. T., Lackmann, G. M., Goodwin, J., Correia Jr, J., & Harnos, K. (2023). An iterative approach toward development of ensemble visualization techniques for high-impact winter weather hazards: Part I: Product development. Bulletin of the American Meteorological Society, 104(9), E1630-E1648.*

11. Line 250 to 253: The problem of using feature mask, instead of the value of reflectivity, to quantify the persistence of the intensity of the snowband over a specific ground site is that the feature mask actually includes a large range of reflectivity values and reflectivity-derived snow rate. Like what you have shown in Figs. 5a and 5b, how do you quantify the changes in the magnitudes within the strong feature?

*We are not quantifying the magnitudes between features. We are using our area x time fraction metric to avoid the very large uncertainties in converting Z to snow rate. We have explained this further in the following addition to the text (repeated from #7):*

*"The relationship between reflectivity and liquid equivalent snow rate has large uncertainty as it depends on the ice particle shape, density, degree of riming, aggregation, and terminal velocities of the snow particles present. It would not be suitable to use the quantitative values from a single Z-S relationship (e.g. Fujiyoshi et al., 1990; Mitchell et al., 1990; Rasmussen et al., 2003; Matrosov et al., 2008; von Lerber et al., 2017; Wen et al., 2017). We developed our technique based on the method of Krajewski et al. (1992) who used an area-threshold method to estimate mean areal rainfall using radar reflectivity observations. Krajewski et al. found that when there is high uncertainty in the Z-R relationship, the rainfall estimates from the area-threshold method perform significantly better than the estimates from the Z-R relationship. Our area × time fraction method estimates space-time integrals from quantities derived from radar reflectivity spatial patterns thereby avoiding issues that would result from using a highly uncertain transformation from reflectivity to snow rate. Yeh and Colle (2025) compared several different identification methods for identifying snow bands in winter storms and found that object-based methods performed better than methods using an absolute or variable reflectivity thresholds."*

12. Also, is snow rate in Figure 5a non-liquid-equivalent? It needs to be clarified.

*We added the following to the text to clarify (line 157):*

*"Note that the snow rate obtained from this relationship represents an instantaneous rate and is not liquid-equivalent unlike the data reported from the ASOS stations."*

13. Section 3.1: It is good to have your figures and analysis supplemented by the video. I would like to view that scenarios in the lower left (89.1%) and upper right (1.5%) quadrants are consistent in supporting that long-lasting elongated high-reflectivity regions observed by operational ground radars correspond to higher surface snowfall rate measured by ASOS. The the low-area x time faction just occurs much more often than high-area x time fraction.

*Examples from each of the 4 quadrants are provided in the Video Supplement and described in the detailed figure captions. The videos have been retitled to clarify.*

| Old Title | New Title |
|---|---|
| 17 December 2020 area x time fraction example | Heavy snow rate and high feature area x time fraction: 16 December 2020 20:00 UTC to 17 December 2020 20:00 UTC for KALB ASOS station in New York. Corresponds to Figure 9 in Tomkins et al (2025). |
| 27 January 2021 area x time fraction example | Low/moderate snow rate and high feature areas x time fraction: 26 January 2021 12:00 UTC and 27 January 2021 00:00 UTC for ASOS station at KVPD in Rhode Island. Corresponds to Figure 10 in Tomkins et al. (2025) |
| 02 February 2021 area x time fraction example | Low/moderate snow rate and low feature area x time fraction: 02 February 2021 00:00 UTC to 03 February 2021 00:00 UTC for ASOS station at KLEB in New Hampshire. Corresponds to Figure 11 in Tomkins et al. (2025) |
| 26 Jan 2021 area x time fraction example | Heavy snow rate and low feature area x time fraction: 27 January 2021 00:00 UTC to 27 January 2021 18:00 UTC for KORH ASOS station in Massachusetts. Corresponds to Figure 12 in Tomkins et al. (2025) |

14. Differences between Fig. 9 (upper right, 1.5% quadrant) and Fig 10 (lower right 4.6% quadrant), is it possible to provide the comparison of the histograms of the reflectivity values that were included in your feature area x time fraction analysis? As the feature mask may have masked out the changes of reflectivity within the feature. Of course, Fig. 9 has slightly larger feature area x time fraction than Fig. 10, 0.62 vs. 0.58, which may have contribution to higher surface snowfall rate too.

*The feature detection algorithm is not a "mask" but an object identification. Below we show data from three examples that we present in Tomkins et al. (2024) to create histograms of the reflectivity and snow rate values within features vs. background.*

*The 3 examples: from Tomkins et al. (2024) refer to their Fig. 3a-f and Fig. 8g-i*

[Figure]

*Caption: Close-up examples of (a)–(c) reflectivity [dBZ] re-scaled to (d)–(f) snow rate [mm h–1] and (g)-(i) feature detection output. Panels (a), (d), and(g) show data from 7 February 2021 14:37:28 UTC with an area of 263 km×222 km; panels (b), (e), and (h) show data from 17 December 2020 16:26:01 UTC with an area of 472 km×361 km; and panels (c), (f), and (i) show data from 17 December 2019 16:23:59 UTC with an area of 326 km×306 km. Grid spacing is 2 km×2 km for all examples.*

[Figure]

*Histograms of the reflectivity and snow rates within features as compared to echo background:*

*a) Distribution of reflectivity within identified features as compared to reflectivity in the background echo, The blue histograms are for background pixels (teal regions in feature detection field) and red histograms are for feature fields (yellow and orange regions in feature detection field). b) and c) show the distributions of snow rates within identified features and background echo. Note that b) has a trimmed y-axis and c) has the full y-axis. The blue box in c) indicates the extent of the graph in b).*

*These distributions illustrate that there is overlap between the reflectivity values in the background echo and in the feature echo. A threshold-based method would miss many weaker locally enhanced features as well as underestimate the size of stronger features.  The two modes in the feature distributions are related to the weak versus strong features.*

15. As you pointed out in the last scenario (Fig. 12), the radar beam height may be above the locally enhanced features. It is very likely that the height of the radar beam, i.e., the distance between the radar site and the ground station also plays a role.

*We tested the sensitivity of the results to the beam height (discussed briefly in Section 3.1.1 and fully in Section 5.1.4 in Tomkins (2024)) and found that thresholding the results by the beam height did not change the findings. The relevant figure illustrating this sensitivity test is below:*

[Figure]

**Figure 5.14:** Sensitivity of results to number of hours of accumulation. 2D distribution of feature area × time fraction versus liquid equivalent precipitation rate [mm hr$^{-1}$] for snow observations accumulated over (a) 2 hours and (b) 3 hours. Area × time fraction calculated with a 25 km radius and and observations are paired with a 0-hour lag. Zero feature area × time fraction observations are removed. 0.5 area × time fraction is annotated with a vertical black dashed line and 7.5 mm hr$^{-1}$ is annotated with a horizontal black dashed line.

---

## Author Response (AR2)

Thanks for the revision. It answered most of my questions. I have two more comments regarding being specific in the key findings and adding the ratio between the actual snowfall rate vs. the liquid equivalent. It is important to be specific about the two details, otherwise the readers will find it confusing.

Comments:
Line 80-82: key findings. Please add two words to specify the 'surface snow rates'. Change it to 'hourly liquid-equivalent surface snow rates'. It is important to be clear in the key findings that the surface liquid-equivalent snow rate you are talking about is from the hourly surface accumulation data (ASOS).

***We have changed the sentence on line 80 to read:***
***"The key finding is that locally enhanced linear features (i.e. mesoscale snow bands) in operational scanning radar reflectivity within northeast US snow storms (which exclude orographic and lake effect snow storms) are usually not associated with heavy hourly liquid-equivalent surface snow rates."***

Line 158: If the S is not liquid-equivalent. You need to provide the information of the approximate scale between liquid-equivalent and non-liquid-equivalent. Is it 10? But for dry snow vs. wet snow, their scales are different. Please provide a range of the scale that is used in the literature and operational forecasting.

***Thank you for pointing this out. The snow rate rescaled from reflectivity is liquid-equivalent, it just isn't hourly. We have changed the sentence beginning on line 157 to read:***
***"Note that the snow rate obtained from this relationship represents an instantaneous liquid equivalent snow rate from a single radar scan and hence is not expected to match the hourly accumulation liquid equivalent reported from the ASOS stations."***

***We have also revised the figure captions in Figs. 4 and 5 to clarify that this is liquid-equivalent snow rate.***

Not sure why the supplement video (for Fig.10) appears not match the Fig 10 in the manuscript.

**Thank you for noticing this. The titles of the videos have been fixed.**